# Finding an Unsupervised Image Segmenter in Each of Your Deep Generative Models

**Luke Melas-Kyriazi**          Christian Rupprecht          Iro Laina          Andrea Vedaldi

University of Oxford
{lukemk,chrisr,iro,av}@robots.ox.ac.uk

## Abstract

Recent research has shown that numerous human-interpretable directions exist in the latent space of GANs. In this paper, we develop an automatic procedure for finding directions that lead to foreground-background image separation, and we use these directions to train an image segmentation model without human supervision. Our method is generator-agnostic, producing strong segmentation results with a wide range of different GAN architectures. Furthermore, by leveraging GANs pretrained on large datasets such as ImageNet, we are able to segment images from a range of domains without further training or finetuning. Evaluating our method on image segmentation benchmarks, we compare favorably to prior work while using neither human supervision nor access to the training data. Broadly, our results demonstrate that automatically extracting foreground-background structure from pretrained deep generative models can serve as a remarkably effective substitute for human supervision.

## 1    Introduction

Recent years have seen rapid progress in the field of deep generative modeling of images, driven by a proliferation of research into Generative Adversaial Networks (GANs) (Goodfellow et al., 2014). Nowadays, it is possible to generate high-resolution images of realistic objects and scenes (Karras et al., 2019; Brock et al., 2019). However, with the exception of generation for computer graphics, there has been limited research into how we might be able to leverage the representations learned by these powerful generative models to enhance other tasks, particularly those involving semantic understanding (e.g. classification or segmentation).

Generally, deep generative models learn to map latent codes to images, imposing simple statistical structures on the distribution of the latent codes, such as assuming an i.i.d. Gaussian distribution. Due to this structure, in some cases code dimensions acquire specific meanings which can be related to human-interpretable concepts (e.g., the rotation or size of an object); however, the code space in high-quality generators (e.g., BigGAN (Brock et al., 2019), BigBiGANs (Donahue & Simonyan, 2019), StyleGAN (Karras et al., 2019)) is usually not easily interpretable. Nonetheless, it is intuitive that an efficient generative process should account for the structure of natural images, including for example the fact that images often comprise distinct foreground and background regions.

In this paper, we validate this hypothesis by learning to separate foreground and background image regions from generator networks. Our approach starts from an arbitrary, off-the-shelf high-quality generator network trained on a large corpus of (unlabeled) images. While these generator networks are *not* explicitly trained for foreground/background segmentation, we show that such a separation emerges *implicitly* as a step to efficiently encode realistically-looking images. Specifically, we design a probing scheme that can extract such foreground/background information *automatically*, i.e. without manual supervision, from the generated images.

This scheme works as follows (cf. Fig. 1). We start from a random code in latent space and learn a fixed, global offset that results in a change in the generated images. The offset is learned to alter the appearance of foreground and background such that a mask can be extracted from the changes in image space.

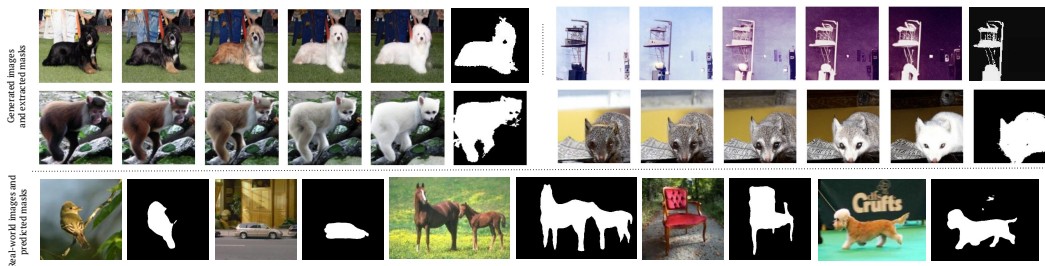

Figure 1: We automatically find a universal latent direction in a GAN that can separate the foreground from the background without any supervision. We can then generate unlimited samples with masks to train a segmentation network that achieves state-of-the-art unsupervised segmentation performance.

The resulting masks provide segmentation maps for the generated images, but they cannot yet be used to segment images from the real world. Given a natural image, the obvious approach would be to find the corresponding code in the latent space of the generator, and then obtain a mask with our method. Unfortunately, this inversion process is less than trivial. In fact, recent work provides strong evidence that the expressiveness of GANs is insufficient to encode arbitrary images (Bau et al., 2019), meaning that the inversion problem has no solution in general.

As we aim to build a general-purpose segmentation method, we take a different approach: we *generate* a labelled image dataset with foreground/background segmentations and use the generated dataset to train a standard segmentation network. With this, we show that our method can successfully learn accurate foreground-background segmentation networks with no manually provided labels at all. Differently from prior work in GAN-based image segmentation, we neither design a new GAN architecture specifically for the task of segmentation nor use manual supervision to extract segmentation information from an existing GAN. Thus, we can discover meaningful latent directions for *any* GAN with no need for model-specific manual intervention.

Extensive experiments on five segmentation datasets across twelve different GANs demonstrate the effectiveness and generalizability of our approach. Moreover, by constructing our image segmenters from generator networks trained on a generic large-scale datasets such as ImageNet, our method can learn to generically segment objects from a wide range of visual domains. Specifically, when we apply our generic segmenters to the CUB200 (Welinder et al., 2010) and Oxford Flowers (Nilsback, 2009) datasets, we attain very strong foreground-background results despite not training on this data. Similarly, when we apply our generic segmenters to three saliency detection benchmarks, our method approaches and sometimes even exceeds the performance of supervised and handcrafted saliency detection methods. An analysis of our results also shows that segmentation performance directly correlates with the quality of the underlying GAN, suggesting that foreground/background separation is an important concept in learning generative models.

Finally, we demonstrate that our method may be used as a drop-in replacement for saliency networks for the purpose of learning pixel-wise semantic image representations. These pixel-wise representations can then be clustered to obtain fully unsupervised *semantic* segmentations, extending our method beyond foreground-background segmentation. Thus, we not only demonstrate for the first time that it is possible to obtain semantic segmentations using GANs, but also that this information may be extracted from a wide range of generic GANs trained on general-purpose image datasets.

## 2 RELATED WORK

Below, we describe how our method relates to recent work in interpreting generative models and object segmentation. Due to space constraints, we include additional related work in the Appendix.

**Interpreting Deep Generative Models.** Several works have proposed methods for decomposing the latent space of a generative model into interpretable or disentangled directions. Early work included Beta-VAE (Higgins et al., 2017), which modified the variational ELBO in the original VAE formulation, and InfoGAN (Chen et al., 2016), which maximized the mutual information between a subset of the latent code and the generated data. Later work has sought to disentangle factors of variation by mixing latent codes (Hu et al., 2018), adding additional adversarial losses (Mathieu et al., 2016), and using contrastive learning (Ren et al., 2021).

Our work follows a recent line of research that looks for structure in large, pretrained generative models. Shen & Zhou (2020) perform a direct decomposition of model weights to find disentangled directions, while Peebles et al. (2020) penalize nonzero second-order interactions between different latent dimensions, and Voynov & Babenko (2020) find interpretable directions by introducing an additional reconstruction network.

Differently from the works above, we conduct a deep study of *one* specific type of structure (foreground/background separation) encoded in the latent space. Other works have taken this approach in the context of extracting 3D structure from 2D images; for example, IG-GAN (Lunz et al., 2020) uses a neural renderer to recover 3D (voxel-based) representations of scenes, and GAN2Shape (Pan et al., 2021) exploits viewpoint and lighting variations in generated images to recover 3D shapes.

**Unsupervised Object Segmentation.** Prior work on unsupervised object segmentation can be divided into two categories: those that employ generative models to obtain segmentation masks and those that employ purely discriminative methods such as contrastive learning (Ji et al., 2019a; Ouali et al., 2020). Here, we focus on generative approaches.

Nearly all generative approaches are based on the idea of decomposing the generative process in a layer-wise fashion; in general, the foreground and background of an image are generated separately and then combined to obtain a final image. Specifically, ReDo (Chen et al., 2019) trains a generator to re-draw new objects on top of old objects, and enforces realism through adversarial training. (Bielski & Favaro, 2019) generates a background, a foreground, and a foreground mask separately and composite them together; they prevent degenerate outputs (i.e. the foreground and background being the same) by randomly shifting the foreground relative to the background. Copy-Paste GAN (Arandjelović & Zisserman, 2019) receives two images as input and copies parts of one image onto the other. OneGAN (Benny & Wolf, 2020) learns to simultaneously generate, cluster, and segment images with a combination of GANs, VAEs, and additional encoders. Equivariant Layered GAN (Yang et al., 2021) first trains a new layerwise GAN and then trains a segmentation network on synthetic data. Labels4Free (Abdal et al., 2021) proposes a new layerwise network inspired by StyleGAN for foreground-background segmenation. Common difficulties with these layer-wise approaches above include the challenges of training new GANs and scaling beyond simple datasets (e.g. CUB, Flowers).

One work along different lines is by Voynov et al. (2021), which uses a pretrained BigBiGAN (Donahue & Simonyan, 2019) generator rather than proposing a new layer-wise GAN. Voynov et al. (2021) use the method from Voynov & Babenko (2020) to decompose the latent space into interpretable directions. A recent version of their method also picks out this direction without supervision using the idea that a foreground-background separating direction should be decomposable into two affine operators acting on different sets of pixels.

Our approach is based on generative modeling, but it differs from other approaches in that we seek a *general-purpose* method to find foreground/background structure *implicitly* encoded in standard, non-layer-wise GANs rather than encoding it *explicitly* or searching for it manually. This enables us to leverage any of the numerous existing generators that have already been pretrained on millions of high-resolution images, rather than developing a new GAN architecture for this specific task. Differently from layer-wise approaches, our approach does not require training new GANs and it does not rely on the assumption that the foreground and background of an image are independent. Differently from Voynov et al. (2021), we show through experiments that our method can be applied to a wide range of different GANs and find that it delivers superior performance on four out of five object segmentation and saliency detection benchmarks. Finally, unlike any of these previous works, we demonstrate that our foreground-background segmentation network can be used as a drop-in replacement for saliency networks for the task of learning dense semantic image representations. By clustering these representations, we are able to extend our method from foregound-background segmentation to semantic segmentation.

## 3 METHOD

Let $x \in \mathbb{R}^{3 \times H \times W}$ be a (color) image. A *generator* (network) is a function $G : \mathbb{R}^D \to \mathbb{R}^{3 \times H \times W}$ that maps code variables $z$ to images $x = G(z)$. Optionally, some generative models come with an encoder function $E : \mathbb{R}^{3 \times H \times W} \to \mathbb{R}^D$ which computes an approximate inverse of the generator (i.e. $G(E(x)) \approx x$).

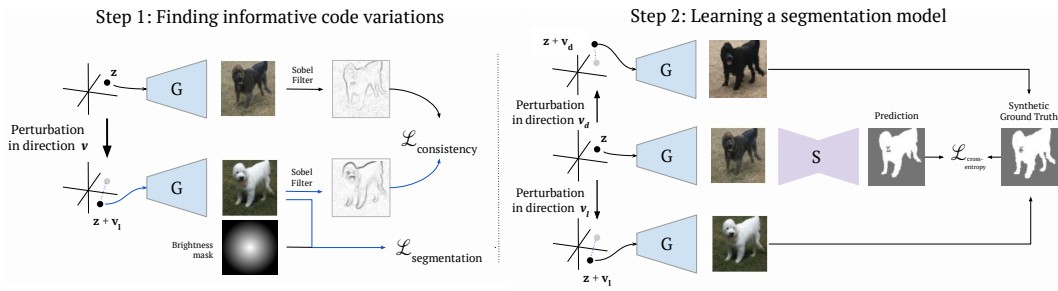

Figure 2: (Typo fixed) Our unsupervised segmentation pipeline. First (left), a direction is identified in the latent space of a deep generative model ($G$) that separates the foreground and background of generated images by changing their relative brightness. Second (right), a synthetic dataset is generated using this direction (or two of these directions) and is used to train a separate segmentation network ($S$). This network can then be applied to unseen real-world data without further training.

A challenge in generating images is that individual pixels exhibit complex correlations, caused by the fact that the images are obtained as the composition of a number of different objects. For example, all pixels that belong to a dog have a similar color, characteristic of dog's instance. However, the correlation is much less strong between pixels that belong to *different* objects. This is because, while object in a scene are not entirely independent, their correlation is much weaker than within the structure of objects.

Intuitively, an image generator must learn to account for such correlations in order to generate realistically-looking images. In particular, we expect the generator to capture the idea that pixels that belong to the same object have a related appearance, whereas the appearance of pixels that belong to different objects or, as it may be, to a foreground object and its background, should be much more statistically independent.

Given a generator function $G$, it is then natural to ask whether such correlations can be extracted and used not just for the purpose of generating images, but also for analyzing them. In order to explore this idea, we consider perturbing the code $z$ via a small increment $\epsilon v \in \mathbb{R}^D$, where $\epsilon \in \mathbb{R}$ and $v \in \mathbb{S}^{D-1}$ is a unit vector. Because the dimension $D$ of the embedding space is typically much smaller than the dimension $3HW$ of the generated images, codes provide highly-compressed views of the data (for example, $D = 120$ for BigBiGAN (Donahue & Simonyan, 2019) and the self-conditioned GAN). As such, most changes in the code are likely to affect most if not all pixels in the image. However, if the generator did in fact learn to compose objects, then one could hope too find specific variations $v$ that only affect only portions of the image, corresponding to individual objects, and use the latter to highlight and segment them.

Empirically, we find that the situation is not as simple. Specifically, it is not easy to find changes in the code that leave part of the pixels exactly constant while changing other pixels. However, we find that there are directions that affect foreground and background regions in a systematic and characteristic manner. Furthermore, we show that these directions are 'universal', in the sense that the *same* $v$ works for *all* codes $z$, and are thus characteristic of a given generator network $G$.

## 3.1 FINDING INFORMATIVE CODE VARIATIONS

Next, we introduce an automated criterion to select informative changes $v$ in code space. To this end, we consider an image $x = G(z)$ generated from a random code $z \sim \mathcal{Z}$, where $\mathcal{Z}$ is the code distribution characteristic of the generator (e.g. an i.i.d. Gaussian). We then consider a modified image $x' = G(z + \epsilon v)$ and observe the change $x \rightarrow x'$.

We compare the two images using two criteria. The first one preserves the *structure* of the image $x$. We capture the latter by imposing that $x$ and $x'$ generate approximately the same edges when fed to a simple edge detector. The intuition is that we wish $v$ to affect the appearance of objects without changing their shape. By preventing objects from 'moving around' the image or deforming, we make it significantly easier to extract an image segmentation from the change $x \rightarrow x'$. This loss

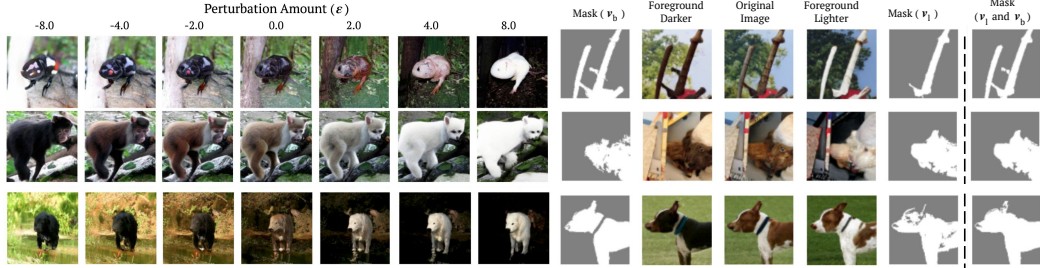

(a) A comparison of generated images for different values of the perturbation length $\epsilon$, using the BigBiGAN generator. A value of $\epsilon = 0.0$ corresponds to the original image, with a random Gaussian latent vector $z \sim \mathcal{N}(0,1)$.

(b) A comparison of perturbed images and their masks for $v_b$ (foreground darker), $v_l$, (foreground lighter), and the combination $v_b$ and $v_l$. Using both directions yields visually superior segmentations.

takes the form:

$$\mathcal{L}_s(v) = \frac{1}{N} \sum_{i=1}^{N} \| S(G(z_i + \epsilon v)) - S(G(z_i)) \|^2$$

where $z_i \sim \mathcal{Z}$ and $S$ is the Sobel-Feldman operator:

$$[S(x)]_{ij} = \sum_{c=1}^{3} (g * x_{c::})_{ij}^2 + (g^\top * x_{c::})_{ij}^2,$$

and $g = \begin{bmatrix} 1 & 2 & 1 \end{bmatrix} \cdot \begin{bmatrix} 1 & 0 & -1 \end{bmatrix}^\top$.

This loss encourages $x$ and $x'$ to be similar. We thus also need a loss that encourages the direction $v$ to explore a non-zero change of the image. We consider an image contrast variation and additionally exploit the photographer bias, that objects are often placed in the middle of the image. This is captured by the loss:

$$\mathcal{L}_c(v) = \frac{1}{N} \sum_{i=1}^{N} \sum_{c=1}^{3} \langle G(z + \epsilon v), r \rangle$$

where $r \in \mathbb{R}^{H \times W}$ is a 'radial' prior:

$$r_{ij} = 1 - \frac{1}{\alpha} \sqrt{ \left( i - \frac{H+1}{2} \right)^2 + \left( j - \frac{W+1}{2} \right)^2 }$$

with normalization factor $\alpha = \frac{1}{4}\sqrt{(H-1)^2 + (W-1)^2}$ that linearly interpolates from 1 in the center of the image to $-1$ at the boundary. This encourages finding a direction $v$ that changes the brightness in the center of the image opposite to the border. In order to learn $v$, the two losses are combined with a weighting factor $\lambda$.

$$\mathcal{L}(v) = \lambda \mathcal{L}_c(v) + \mathcal{L}_s(v) \tag{1}$$

Given this fully-automatic procedure, the latent code direction $v$ may be thought of as a function of the generator $G$ and the weighting factor $\lambda$.

### 3.1.1 COMBINING INFORMATIVE CODES

Optimizing Eq. (1) with $\lambda > 0$ encourages the network to produce a shift $v$ that brightens the foreground and darkens the background of an image. However, there is no constraint that $\lambda$ need be positive; by negating $\lambda$ and optimizing a second time, we obtain another direction $v$ that shifts the foreground dark and the background light.

Although using only one direction suffices for our method, we find that we can improve performance by using both. As a result, for the remainder of the paper, let $v_l$ represent the direction that shifts the foreground lighter, and $v_d$ represent the direction that shifts the foreground darker.

## 3.2 LEARNING A SEGMENTATION MODEL

Once the latent directions $v_d$ and $v_l$ have been found, the process of extracting a segmentation mask is straightforward: we label as foreground regions the pixels in which the image generated with the foreground-lighter shifted latent code is lighter than the image generated with the foreground-darker shifted latent code. That is, for a generated image $x = G(z)$, we have:

$$M(z) = \text{sign}(G(z + \epsilon v_l) - G(z + \epsilon v_d)) \tag{2}$$

Alternatively, if we use only a single direction $v_l$ or $v_d$, $M(z)$ is set to either:

$$G(z + \epsilon v_l) - G(z) \quad \text{or} \quad G(z) - G(z + \epsilon v_d) \tag{3}$$

Given the learned direction $v$, we use it to generate a training set as follows:

$$\mathcal{D} = \{(G(z_i), M(z_i)) : z_i \sim \mathcal{Z}, i = 1, \ldots\}.$$

This dataset may then be used to train any dense segmentation network $\Psi$ (i.e., a UNet (Ronneberger et al., 2015)) in the standard fashion. That is, we minimize the pixel-wise binary cross-entropy loss between the segmentation output $\Psi(G(z)) \in \mathbb{R}^{H \times W}$ and the (synthesized) mask $M(z)$:

$$\mathcal{L}(\Psi|z) = -\frac{1}{HW} \sum_{u \in [H] \times [W]} \log p(\text{sign}(M_u(z))|\Psi_u(G(z)))$$

where $p(m|s) = m\sigma(s) + (1-m)\sigma(-s)$, $u$ is a pixel index and sign is the sign function. Unlike previous object segmentation methods, our method requires no additional losses or constraints to ensure the stability of training. By sampling $z$, we can generate an 'infinite' dataset for learning the network $\Psi$. Although we described the procedure above for unconditional GANs, our method applies just as well for weakly-supervised conditional GANs, where the generator $G(z, y)$ also depends on a class label; we simply sample a label $y$ uniformly at random for each generated image.

### 3.2.1 REFINING THE GENERATED DATASET

An advantage of training with GAN-generated data is that the dataset size is infinite, which means that one is free to curate one's dataset and discard uninformative training examples. In our case, we found that it was helpful to refine the dataset by (1) discarding images with masks that were too large, (2) discarding images for which the latent code shift did not produce a significant change in brightness, and (3) removing small connected components from the mask. The exact details are given in the Supplementary Material.

## 4 EXPERIMENTS

In this section, we present an extensive set of experiments demonstrating the method's performance, its wide applicability across image datasets, and its generalizability across GAN architectures.

### 4.1 EXPERIMENTAL SETUP

As our method is generator-agnostic, we apply our method to twelve generators, including three unconditional and nine conditional GANs. For the three unconditional GANs (BigBiGAN (Donahue & Simonyan, 2019), SelfCondGAN (Liu et al., 2020), and UncondGAN (Liu et al., 2020)), our procedure is completely unsupervised. For conditional GANs, our method is still unsupervised but the GAN naturally relies on class supervision for training.

To demonstrate the efficacy of our method across resolutions and datasets, we implement both, GANs trained on ImageNet (Deng et al., 2009) at a resolution of 128px, and GANs trained on the smaller TinyImageNet dataset (100,000 images split into 200 classes) at a resolution of 64px. All experiments performed across all GANs utilize the same set of hyperparameters for both optimization and segmentation. This is a key advantage of our method relative to other unsupervised/weakly-supervised image segmentation methods (Chen et al., 2019; Bielski & Favaro, 2019; Benny & Wolf, 2020; Arandjelović & Zisserman, 2019), which are sensitive to dataset-specific hyperparameters.

Table 1: (Updated for Voynov et al. (2021)) Performance on three saliency detection benchmarks (DUTS, EC-SSD, DUT-OMRON) and two object segmentation benchmarks (CUB, Flowers). ** initializes with a pretrained supervised network. † CRF post-processing. ◇ our implementation.

| | DUTS | | | ECSSD | | |
|---|---|---|---|---|---|---|
| | Acc | IoU | $F_\beta$ | Acc | IoU | $F_\beta$ |
| *Supervised Methods* | | | | | | |
| (Hou et al., 2019) | 0.924 | - | 0.729 | 0.930 | - | 0.880 |
| (Luo et al., 2017) | 0.920 | - | 0.736 | 0.934 | - | 0.891 |
| (Zhang et al., 2017b) | 0.902 | - | 0.693 | 0.939 | - | 0.883 |
| (Zhang et al., 2017c) | 0.868 | - | 0.660 | 0.920 | - | 0.852 |
| (Wang et al., 2017) | 0.915 | - | 0.672 | 0.908 | - | 0.826 |
| (Li et al., 2016) | 0.924 | - | 0.605 | 0.840 | - | 0.759 |
| *Handcrafted Methods* | | | | | | |
| RBD (Zhu et al., 2014) | 0.799 | - | 0.510 | 0.817 | - | 0.652 |
| DSR (Li et al., 2013) | 0.863 | - | 0.558 | 0.826 | - | 0.639 |
| MC (Jiang et al., 2013) | 0.814 | - | 0.529 | 0.796 | - | 0.611 |
| HS (Zou & Komodakis, 2015) | 0.773 | - | 0.521 | 0.772 | - | 0.623 |
| *Deep Ensembles of Handcrafted Methods* | | | | | | |
| SBF (Zhang et al., 2017a) | 0.865 | - | 0.583 | 0.915 | - | 0.787 |
| USD** (Zhang et al., 2018) | 0.914 | - | 0.716 | 0.930 | - | 0.878 |
| USPS**† (Nguyen et al., 2019) | 0.938 | - | 0.736 | 0.937 | - | 0.874 |
| *Unsupervised Methods* | | | | | | |
| (Voynov et al., 2021) | 0.878 | 0.498 | - | 0.899 | 0.672 | - |
| (Voynov et al., 2021) ◇ | 0.881 | 0.508 | 0.600 | 0.906 | 0.685 | 0.790 |
| Ours | 0.893 | 0.528 | 0.614 | 0.915 | 0.713 | 0.806 |

| | CUB | | | Flowers | | |
|---|---|---|---|---|---|---|
| | Acc | IoU | max$F_\beta$ | Acc | IoU | max$F_\beta$ |
| *Unsupervised Methods* | | | | | | |
| PertGAN (Bielski & Favaro, 2019) | - | 0.380 | - | - | - | - |
| ReDO (Chen et al., 2019) | 0.845 | 0.426 | - | 0.879 | 0.764 | - |
| WNet† (Xia & Kulis, 2017) | - | 0.248 | - | - | - | - |
| UISB (Kanezaki, 2018) | - | 0.442 | - | - | - | - |
| IIC-seg (Ji et al., 2019b) | - | 0.365 | - | - | - | - |
| OneGAN (Benny & Wolf, 2020) | - | 0.555 | - | - | - | - |
| (Voynov et al., 2021) | 0.930 | 0.683 | 0.794 | 0.765 | 0.540 | 0.760 |
| (Voynov et al., 2021)◇ | 0.931 | 0.693 | 0.807 | 0.777 | 0.529 | 0.672 |
| Ours (BigBiGAN - ImageNet) | 0.921 | 0.664 | 0.783 | 0.796 | 0.541 | 0.723 |

| | DUT-OMRON | | |
|---|---|---|---|
| | Acc | IoU | $F_\beta$ |
| (Voynov et al., 2021)* | 0.856 | 0.453 | - |
| (Voynov et al., 2021)*◇ | 0.859 | 0.460 | 0.533 |
| Ours | 0.883 | 0.509 | 0.583 |

Table 2: A comparison of segmentation model performance across a wide range of generator architectures, using a foreground-lighter shift ($v_l$). All hyperparameters are kept constant across generators. IN-128px refers to ImageNet at resolution 128px, and TinyIN-64px refers to TinyImageNet at resolution 64px.

| | | CUB | | Flowers | | DUT-OMRON | | DUTS | | ECSSD | |
|---|---|---|---|---|---|---|---|---|---|---|---|
| | Dataset Res. | Acc | IoU | Acc | IoU | Acc | IoU | Acc | IoU | Acc | IoU |
| *ACGAN* (Odena et al., 2017) | TinyIN 64px | 0.682 | 0.265 | 0.572 | 0.266 | 0.642 | 0.190 | 0.647 | 0.191 | 0.652 | 0.276 |
| *BigGAN* (Brock et al., 2019) | TinyIN 64px | 0.853 | 0.257 | 0.723 | 0.284 | 0.844 | 0.213 | 0.842 | 0.224 | 0.811 | 0.332 |
| *GGAN* (Lim & Ye, 2017) | TinyIN 64px | 0.818 | 0.366 | 0.697 | 0.315 | 0.782 | 0.221 | 0.783 | 0.235 | 0.766 | 0.316 |
| *SAGAN* (Zhang et al., 2019) | TinyIN 64px | 0.828 | 0.376 | 0.732 | 0.351 | 0.808 | 0.235 | 0.806 | 0.246 | 0.788 | 0.327 |
| *SNGAN* (Miyato et al., 2018) | TinyIN 64px | 0.849 | 0.357 | 0.751 | 0.374 | 0.816 | 0.216 | 0.814 | 0.217 | 0.795 | 0.292 |
| *SAGAN* (Zhang et al., 2019) | IN 128px | 0.871 | 0.336 | 0.608 | 0.085 | 0.856 | 0.250 | 0.860 | 0.282 | 0.814 | 0.340 |
| *SNGAN* (Miyato et al., 2018) | IN 128px | 0.881 | 0.378 | 0.703 | 0.304 | 0.860 | 0.305 | 0.854 | 0.300 | 0.837 | 0.432 |
| *ContraGAN* (Kang & Park, 2020) | IN 128px | 0.857 | 0.159 | 0.661 | 0.088 | 0.858 | 0.075 | 0.870 | 0.149 | 0.805 | 0.204 |
| *UnCondGAN* (Liu et al., 2020) | IN 128px | 0.734 | 0.217 | 0.494 | 0.049 | 0.698 | 0.127 | 0.729 | 0.158 | 0.681 | 0.198 |
| *SelfCondGAN* (Liu et al., 2020) | IN 128px | 0.869 | 0.459 | 0.670 | 0.238 | 0.800 | 0.280 | 0.806 | 0.297 | 0.806 | 0.412 |
| *BigGAN* (Brock et al., 2019) | IN 128px | 0.886 | 0.367 | 0.731 | 0.318 | 0.883 | 0.316 | 0.876 | 0.303 | 0.848 | 0.424 |
| *BigBiGAN* (Donahue & Simonyan, 2019) | IN 128px | **0.912** | **0.601** | **0.773** | **0.479** | **0.878** | **0.451** | **0.890** | **0.486** | **0.905** | **0.663** |

## 4.2 EVALUATION DATA

We evaluate the performance of our model on three standard saliency detection benchmarks (DUT-Omrom (Yang et al., 2013), DUTS (Wang et al., 2017), ECSSD (Shi et al., 2016)) and two standard object segmentation benchmarks (CUB (Welinder et al., 2010), and Flowers-102 (Nilsback & Zisserman, 2009)). For the saliency datasets, we evaluate using (pixel-wise) accuracy, mean intersection-over-union (IoU), and $F_\beta$-score with $\beta^2 = 0.3$. For the object segmentation datasets, we evaluate using accuracy, IoU, and max$F_\beta$ (the maximum $F_\beta$ score over a range of 255 uniformly distributed binarization thresholds between 0 and 1).

## 4.3 RESULTS

**Performance on Benchmarks.** In Table 1, we compare our method to other recent work. We emphasize that our method uses the same model for all datasets and has not seen any of the (training or evaluation) data for these datasets before. Our method delivers strong performance across datasets, approaching and even outperforming some supervised/handcrafted saliency detection methods. In

comparison to Voynov et al. (2021), we perform better on four out of five benchmarks (all except CUB), even though we do not rely on humans to hand-pick latent directions. In comparison to layerwise GANs (Chen et al., 2019; Bielski & Favaro, 2019; Benny & Wolf, 2020; Arandjelović & Zisserman, 2019), we perform similarly on CUB and Flowers-102, but we cannot compare our method with layerwise GANs on complex datasets (e.g. DUTS) because they do not produce any meaningful results. Due to the difficulty of training GANs, they are only ever trained on datasets consisting of images from a single domain with a single main subject, such as birds or flowers. By contrast, our ability to leverage pre-trained generators means that our method scales to complex and diverse datasets, such as those used for saliency detection.

**Performance across Generators.** We investigate the generality of our method by performing the same optimization and training pipeline with twelve different GANs. For each generator, we optimize to obtain a latent direction $v_l$, train a segmentation model using this direction, and evaluate its performance across the five datasets above. The same hyperparameters are kept constant for *all* GANs, including $\lambda = 5.0$ during the optimization phase.

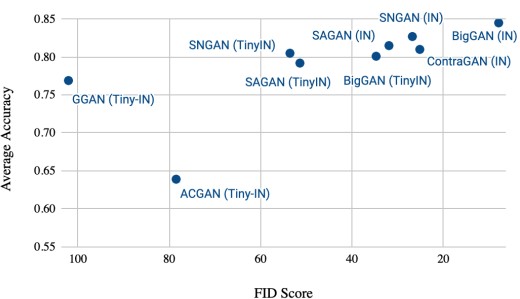

Figure 4: A plot of Frechet Inception Distance (FID) versus average segmentation accuracy across all five evaluation datasets (CUB, Flowers, DUT-OMRON, DUTS, ECSSD) for nine different GAN architectures. Lower FIDs are better (note that the x-axis is reversed). Lower FID scores correlate with improved final segmentation accuracy.

Results are shown in Table 2; BigBiGAN performs best, but all networks, even those using relatively weak TinyImageNet-trained GANs (e.g., GGAN (Lim & Ye, 2017)), deliver reasonable segmentation performance. This highlights the benefits of our fully-automatic segmentation pipeline; our method performs well across a wide range of generators trained on different datasets at different resolutions.

Naturally, the quality of a final segmentation network produced by our method is related to the quality of the underlying generator. Figure 4 plots the Frechet Inception Distance (FID) score of nine conditional GANs versus the average accuracy of the corresponding segmentation networks produced by our method. Lower FID scores, which correspond to better GANs, correlate with improved accuracy. This correlation suggests that as better GANs architectures are developed, our method will continue to produce better unsupervised segmentation networks.

**Ablation: Comparing latent directions.** We compare the performance of a segmentation masks using the two latent directions $v_d$ and $v_l$ together (Eq. (2)), or each of them individually (Eq. (3)) visually in Fig. 3a. In Table 6, we quantitatively compare the results of these three methods along with a fourth method in which we ensemble the final segmentation networks produced by $v_d$ and $v_l$ individually. The foreground-lighter ($v_l$) and foreground-darker ($v_d$) directions yield similar results when used individually. The combination ($v_l$ and $v_d$) provides superior results, on par with the ensemble. Unlike the ensemble, which requires training two networks, the combination of $v_d$ and $v_l$ adds minimal overhead compared to training with one direction.

**Ablation: Varying $\lambda$, $\epsilon$, the Central Prior, Random Initializations, Amount of Generated Data.** The two hyperparameters in the optimization stage of our method are $\lambda$, which controls the trade-off between brightness and consistency, and $\epsilon$, which controls the magnitude of the perturbation. We find that the process is only modestly sensitive to changes in these hyperparameters. We also find that using the central prior compared to a spatially-agnostic loss term is better, but only moderately. Detailed numerical results, including the ablation on the central prior, are given in Section A.3. In Appendix B.3, we also investigate how segmentation performance varies with the number of generated images, finding a clear log-linear relationship.

**Qualitative Results.** By inspection, we find that our optimization procedure is able to edit images in such a manner that the foreground becomes lighter and the background becomes darker. In numerous cases, the network appears to convert the scene from daytime to nighttime. Furthermore, better GANs generally produce qualitativley better segmentations. Please refer to Appendix A and Section A.1.3 for illustrations.

Table 3: A comparison of semantic segmentation performance on Pascal-VOC obtained from $K$-means clustering of pixelwise semantic features. The mIoU is computed over the 20 classes by performing Hungarian matching between the clusters obtained from $K$-means and the ground truth. All networks use a ResNet backbone. We compare with numerous baselines, including using self-supervised features directly (i.e. MoCo, SwaV) and IIC. Compared to Van Gansbeke et al. (2021), we achieve competitive performance, but our pipeline is entirely unsupervised, whereas theirs uses a saliency network which was initialized with a supervised network pretrained for semantic segmentation on CityScapes.

|  | Method | Saliency Network | Saliency Net. PT | Sem. Seg. PT | mIoU |
|---|---|---|---|---|---|
| Colorization | Proxy task | - | - | Colorization | 4.9 |
| IIC | Clustering | - | - | IIC | 9.8 |
| MoCo | Image Contrast | - | - | Moco | 4.3 |
| Swav | Image Contrast | - | - | Swav | 4.4 |
| ImageNet Sup. | Image Contrast | - | - | Sup. ImageNet | 4.4 |
| MaskContrast | Pixel Contrast | DeepUSPS + BAS-Net | Cityscapes (Sup.) | MoCo | 35.0 |
| MaskContrast | Pixel Contrast | DeepUSPS + BAS-Net | Cityscapes + DUTS (Sup.) | MoCo | **38.9** |
| Ours (BigBiGAN) | Pixel Contrast | Our method | Our method (Unsup.) | MoCo | *36.5* |

## 4.4 EXTENSION TO SEMANTIC SEGMENTATION

Finally, we demonstrate that our network may be extended from binary foreground-background segmentation to semantic segmentation, the task of assigning each pixel in an image into one of $K$ semantic categories. Due to the challenging nature of this task, it has not been attempted by any previous works in the GAN-based segmentation space.

We extend our method by following the dense contrastive learning approach proposed by Van Gansbeke et al. (2021). In this approach, binary masks are extracted from a set of images using a foreground-background segmentation model, sometimes called a "mid-level visual prior." Then, a network is trained to generate pixel-wise features using a mask-based contrastive loss: features corresponding to pixels in the foreground of the image are pulled toward the features of other pixels in the mask and pushed away from the features of background pixels. After training, semantic segmentations may be extracted by clustering these pixel-wise features across an entire dataset.

Importantly, Van Gansbeke et al. (2021) uses saliency detection networks to generate their object masks. Although these saliency detectors are sometimes called "unsupervised," they are actually initialized using pretrained semantic segmentation networks[1]. We propose to use our object segmentation network as a drop-in replacement for these saliency networks, making the entire process entirely unsupervised. We leave the rest of their method (i.e. the dense contrastive learning and the evaluation procedure) unchanged.

We perform experiments on the PASCAL VOC dataset, which contains 20 semantic classes. Experimental details are included in Section A.1.3 and K-Means clustering results are shown in Table 3. We compare to a range of baselines, along with two models from Van Gansbeke et al. (2021) using different levels of supervised pretraining. Our network is competitive with Van Gansbeke et al. (2021) despite being entirely unsupervised. Furthermore, in Table 4 in Section A.1.3 we show that this procedure works for a wide range of GANs, demonstrating the generalizability of our approach.

## 5 CONCLUSIONS

We find that extracting a salient object segmentation from the latent space of a GAN is not only possible without supervision but also leads to state-of-the-art unsupervised segmentation performance on several benchmark datasets. In contrast to existing methods that have been engineered specifically for this task, we extract segmentations from a network trained for a very different purpose — generating images. Surprisingly, we are able to generalize to a wide range of segmentation benchmarks without directly training on any real images, and even extend our results to semantic segmentation. Our findings directly prompt future research questions about what other concepts of the physical world can be automatically extracted from generative models, and to what extent we can use such extracted concepts to replace human supervision in other computer vision tasks.

---

[1]In the case of Van Gansbeke et al. (2021), they use DeepUSPS(Nguyen et al., 2019), which is initialized using a supervised network pretrained for semantic segmentation on Cityscapes.

## 6 REPRODUCIBILITY STATEMENT

We aim to ensure that our experiments are entirely and easily reproducible. We upload code to the Supplementary Material to fully reproduce all experiments. This code contains a README file with a detailed description of the code structure, which should help enable others to reproduce and later extend upon our work. We also take care to describe all hyperparameters and implementation details in the Appendix. Our results do not require extremely large amounts of compute; they can be reproduced with a single GPU by researchers with computational constraints.

## 7 ETHICS STATEMENT

It is important to discuss the potential ethical issues involved with training large-scale generative models and segmentation networks along the lines proposed by our paper.

First of all, the task of segmentation is predicated upon classifying objects into predetermined (either binary or semantic) categories; the definition of these categories, especially in the case of semantic segmentation, may introduce biases into the task itself. Second, when training models on large-scale datasets, it is essential to consider the biases and data privacy issues introduced in the data collection process. For example, the PASCAL-VOC dataset, which we use for the task of semantic segmentation, is composed of images scraped from Flickr. As a result, it is composed primarily of photographs from the United States and Europe, and the "person" class contains primarily images of white individuals. Additionally, it is not clear whether the individuals in these photographs consent to being used to train image segmentation models, bringing up the issue of data privacy.

From an ethical perspective, our method is slightly different from standard segmentation models because it is trained solely on GAN-generated images; this is not to say that it is ethically better or worse, but that it involves different ethical considerations. On the one hand, this might alleviate some data privacy concerns, as the segmentation training data is synthetic. However, since this training data is generated by a GAN, one has to examine the data and methodology originally used to pretrain the GAN; any biases present in this data will likely be reproduced or amplified by the GAN . For example, if one uses a GAN trained on ImageNet to perform object segmentation, it may perform better on white individuals than individuals of other races due to the disproportionate percentage of white individuals in the training data (Steed & Caliskan, 2021). Investigating biases introduced by GANs remains an active area of research in the machine learning community (Jain et al., 2020; Tan et al., 2020), and these ethical discussions extend to our GAN-based segmentation method.

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

# A APPENDIX

## A.1 IMPLEMENTATION DETAILS

### A.1.1 OPTIMIZATION

First, we optimize for the latent code vectors $v_d$ and $v_l$. We generate latent codes $z \sim \mathcal{N}(0,1)$ and optimize the vector $v_l$ (or $v_d$) by gradient descent with the Adam (Kingma & Ba, 2014) optimizer and learning rate 0.05. We use $\lambda = 5$ for the light direction $v_l$ and $\lambda = -5$ for the dark direction $v_b$. We perform 1000 optimization steps, by which point $v_l$ (or $v_d$) has converged.

### A.1.2 OBJECT SEGMENTATION

To generate data, we sample $z \sim \mathcal{N}(0,1)$, produce the mask $m = M(z)$, and refine the mask as described in the main paper. We use a simple UNet (Ronneberger et al., 2015) with bilinear down/up-sampling as our segmentation network. Following (Voynov et al., 2021), we train for 12000 steps using Adam with learning rate $10^{-3}$ and batch size 95, decaying the learning rate by a factor of 0.2 at iteration 8000.

### A.1.3 SEMANTIC SEGMENTATION

We follow the procedure from (Van Gansbeke et al., 2021), but we use binary masks produced by our object segmentation model as a drop-in replacement for their saliency detection masks. Under this procedure, we begin by applying our segmentation network to extract masks from the `trainaug` split of PASCAL-VOC (10582 images). We then train a dilated ResNet-50 model with DeepLab-v3 head, initialized with a pretrained MoCo ResNet-50. As in (Van Gansbeke et al., 2021), this network has a main head and an auxiliary head: the main head predicts dense pixel-wise features of dimension 32, and the auxiliary head predicts the object segmentation mask for the purpose of regularization. We train the output of the main head using a pixel-wise version of the InfoNCE loss. Specifically, for a feature $z_i$ corresponding to a pixel $i$ in the foreground mask $M_i$, we have:

$$\mathcal{L}_i = -\log \frac{\exp(z_i \cdot z_+/\tau)}{\sum_{k=0}^{K} \exp(z_i \cdot z_-/\tau)}$$

where $z_+$ is the average embedding of other pixels in the mask $M_i$, and $z_-$ is the average embeding of pixels in the background:

$$z_+ = \frac{1}{|M_i|} \sum_{i \in M_i} z_i, \qquad z_- = \frac{1}{N - |M_i|} \sum_{i \notin M_i} z_i$$

where $N$ is the number of pixels in the image.

We use the same hyperparameters as (Van Gansbeke et al., 2021) for the contrastive learning procedure. We train for 40 epochs using random crops of size 224, learning rate 0.01, and output dimension 32. After training, we evaluate with K-Means clustering with 21 clusters (20 semantic categories and 1 background category). These clusters are computed globally over the entire validation set and matched to the ground truth labels using the Hungarian algorithm.

We repeat this same procedure identically for a large range of GANs. Figure 9 compares the result of final segmentation networks for different GANs: all images produce sensible segmentations, with better GANs generally producing better segmentations.

## A.2 DATASETS

Below, we give statistics of our evaluation datasets. Note that these are only used for evaluation, as only GAN-generated images are seen during training:

| GAN | Dataset | Resolution | mIoU |
|---|---|---|---|
| BigBiGAN | ImageNet | 128px | 0.345 |
| SAGAN | ImageNet | 128px | 0.259 |
| SelfCondGAN | ImageNet | 128px | 0.228 |
| UncondGAN | ImageNet | 128px | 0.157 |
| ContraGAN | ImageNet | 128px | 0.153 |
| BigGAN | Tiny-ImageNet | 64px | 0.124 |
| SNGAN | Tiny-ImageNet | 64px | 0.095 |

Table 4: Semantic segmentation performance on Pascal-VOC obtained from $K$-means clustering of pixelwise semantic features. mIoU is computed over the 20 classes by performing Hungarian matching between the clusters obtained from $K$-means and the ground truth. We see that, as with object segmentation, better GANs (i.e. BigBiGAN, SAGAN) yield better downstream semantic segmentation performance.

| | CUB | | Flowers | | DUT-OMRON | | DUTS | | ECSSD | |
|---|---|---|---|---|---|---|---|---|---|---|
| | Acc | IoU | Acc | IoU | Acc | IoU | Acc | IoU | Acc | IoU |
| $v_l$ only | 0.912 | 0.601 | 0.773 | 0.479 | 0.878 | 0.451 | 0.890 | 0.486 | 0.905 | 0.663 |
| $v_d$ only | 0.912 | 0.631 | 0.806 | 0.572 | 0.842 | 0.442 | 0.864 | 0.478 | 0.899 | 0.672 |
| $v_l$ and $v_d$ | **0.921** | **0.664** | 0.796 | 0.541 | **0.883** | **0.509** | 0.893 | **0.528** | 0.915 | **0.713** |
| Ensemble | **0.921** | 0.650 | **0.799** | **0.544** | 0.881 | 0.492 | **0.894** | 0.524 | **0.917** | **0.713** |

Table 6: A comparison of segmentation performance when different directions in the latent space are used to construct the training segmentation masks.

| Dataset | Num. Images | Type | Crop |
|---|---|---|---|
| CUB | 1000 | Object seg. | ✓ |
| Flowers | 1020 | Object seg. | ✓ |
| OMRON | 5168 | Saliency det. | ✗ |
| DUTS | 5019 | Saliency det. | ✗ |
| ECSSD | 1000 | Saliency det. | ✗ |

Table 5: Evaluation dataset statistics

## A.3 ADDITIONAL ABLATIONS

**Ablation: Comparing latent directions.** In addition to comparing the networks resulting from $v_l$ and $v_b$, we also compare the actual latent directions $v_l$ and $v_b$. Due to the nonlinearity of the generator function, the optimal unit directions $v_l$ and $v_d$ are not necessarily negations of one another; indeed, we found in practice that they were close to but not exactly antiparallel. Table 9 in Section A.3 gives exact numbers for a variety of different generator architectures.

**Ablation over $\lambda$.** The most notable hyperparameter in the optimization stage of our method is $\lambda$, which controls the trade-off between brightness and consistency. Table 7 compares segmentation results for BigBiGAN with $\lambda = 10, 5, 2.5$, and $1.25$, with $5$ performing best.

**Ablation over $\epsilon$.** In Table 8, we show ablation results for changing $\epsilon$ during the optimization process. Note that since the GAN used in this set of experiments (BigBiGAN) has a 120-dimensional latent space, the distribution of the norm of the $\mathcal{N}(0,1)$ latent vectors used to train the GAN is concentrated around (approximately) 11. That is to say, a shift of magnitude $\epsilon = 6$ in the latent space is very large.

**Ablation: Random initialization.** Given the importance of the latent direction optimization procedure in our segmentation pipeline, it is natural to ask whether different initializations yield materially different directions in the latent space. To answer this question, we optimized ten vectors $v$ on

|  | CUB | | Flowers | | DUT-OMRON | | DUTS | | ECSSD | |
|---|---|---|---|---|---|---|---|---|---|---|
|  | Acc | IoU | Acc | IoU | Acc | IoU | Acc | IoU | Acc | IoU |
| $\lambda = 10$ | 0.911 | 0.631 | 0.794 | 0.550 | 0.849 | 0.455 | 0.874 | 0.498 | 0.899 | 0.677 |
| $\lambda = 5$ | **0.919** | **0.658** | **0.782** | **0.506** | **0.880** | **0.498** | **0.891** | **0.524** | **0.912** | **0.703** |
| $\lambda = 2.5$ | 0.818 | 0.418 | 0.728 | 0.456 | 0.762 | 0.311 | 0.765 | 0.311 | 0.792 | 0.467 |
| $\lambda = 1.25$ | 0.791 | 0.385 | 0.713 | 0.449 | 0.740 | 0.296 | 0.743 | 0.296 | 0.771 | 0.446 |

Table 7: A comparison of segmentation performance for a BigBiGAN model when different values of $\lambda$ are used to find the latent vectors $v_l$ and $v_b$ in the optimization stage. Higher values of $\lambda$ yield latent directions $v$ that produce shifted images with greater variance in brightness between the center and outside pixels. Conversely, lower values of $\lambda$ yield latent directions $v$ that produce shifted images that align better to the original images.

|  | CUB | | Flowers | | DUT-OMRON | | DUTS | | ECSSD | |
|---|---|---|---|---|---|---|---|---|---|---|
|  | Acc | IoU | Acc | IoU | Acc | IoU | Acc | IoU | Acc | IoU |
| $\epsilon = 1$ | 0.911 | 0.600 | 0.744 | **0.600** | 0.867 | **0.454** | 0.880 | 0.479 | 0.897 | 0.650 |
| $\epsilon = 2$ | **0.912** | **0.601** | **0.773** | 0.479 | **0.878** | 0.451 | **0.890** | **0.486** | **0.905** | **0.663** |
| $\epsilon = 4$ | 0.843 | 0.435 | 0.617 | 0.435 | 0.763 | 0.290 | 0.775 | 0.297 | 0.779 | 0.419 |
| $\epsilon = 6$ | 0.761 | 0.347 | 0.602 | 0.347 | 0.714 | 0.236 | 0.709 | 0.238 | 0.724 | 0.349 |

Table 8: A comparison of segmentation performance for a BigBiGAN-based model when different values of $\epsilon$ are used to find the latent vector $v_l$ in the optimization stage. Higher values of $\epsilon$ correspond to a greater-magnitude shift in the latent space during optimization.

BigBiGAN from different initializations and measured their pairwise cosine similarities. We found that the cosine similarities had mean 0.9866 with standard deviation 0.0044, which is extremely small relative to the values in Table 9. Thus, the random initialization of the optimization procedure does not have a large impact on the final direction.

**Ablation: On the necessity of the center prior.** Here, we conduct a small experiment to investigate the necessity of the central prior in the optimization step of our segmentation pipeline. We replace it with a *spatially-agnostic* loss term with no central prior: it simply encourages the shifted image to have high variance in brightness. We find that this spatially-agnostic loss still yields good results, although not quite as good as the spatial prior loss. Numerical results are shown in Table 10.

## A.4 ADDITIONAL RELATED WORK

**Saliency Detection.** Object segmentation is closely related to saliency detection, the problem of finding significant (salient) objects in an image. Although most saliency detection models are trained with pixel-level supervision (or human eye movement data), the past few years have seen some research into unsupervised/weakly-supervised saliency detection (Zhang et al., 2018; Nguyen et al., 2019; Zeng et al., 2019). These methods work by ensembling strong hand-crafted priors and distilling them into a deep network. In practice, they also initialize their networks with pretrained (supervised) image classifiers or semantic segmentation networks.

**Learning from Synthetic Data.** Finally, our method can be viewed from the perspective of learning from synthetic data. Motivated by the costly and time-consuming nature of data labeling, there has been a plethora of recent work on synthesizing, training with, and adapting from synthetic datasets. For example, one widely-studied line of research (Hoffman et al., 2018; Tsai et al., 2018; Zou et al., 2018; Vu et al., 2019; Toldo et al., 2020) tackles the task of semantic segmentation by training on data generated from video games (e.g. GTA5). With regard to adversarially-generated training data specifically (Shrivastava et al., 2017) uses a GAN-like network to enhance the realism of synthetic images while preserving label information.

Although we train our segmentation network using generated images only, we show in the experiments below that it generalizes to real-world images without the need for additional adaptation. Such additional adaptation could be an interesting avenue for future research.

| GAN | $v_s^T v_b$ |
|---|---|
| BigBiGAN | -0.4376 |
| SelfCondGAN | -0.7854 |
| UncondGAN | -0.3522 |
| ContraGAN | -0.3297 |
| SAGAN | -0.4648 |

Table 9: The dot product of the optimized foreground-lighter ($v_l$) and foreground-darker latent directions ($v_b$) for different generators, all of which have a 120-dimensional latent space. Across the board, the directions are almost but not exactly antiparallel (random vectors in this space have an expected dot product of 0 with variance $\frac{1}{120}$).

| | CUB | Flowers | DUT-O. | DUTS | ECSSD |
|---|---|---|---|---|---|
| Acc. | 0.885 | 0.788 | 0.825 | 0.849 | 0.876 |
| IoU | 0.535 | 0.553 | 0.379 | 0.416 | 0.609 |
| maxF$_\beta$ | 0.672 | 0.689 | 0.473 | 0.526 | 0.728 |

Table 10: Ablation results for the spatially-agnostic spatial mask

## A.5 ADDITIONAL EXAMPLES

### A.5.1 EXAMPLES ACROSS DATASETS

In Figure 6, Figure 7,aand Figure 8, we show the results of applying our final segmentation network to random images from each of the five datasets on which we evaluated.

### A.5.2 EXAMPLES ACROSS GENERATORS

In Figure 9, we show examples of pairs of generated images and the corresponding extracted samples for a range of different GANs.

### A.5.3 SEMANTIC SEGMENTATION EXAMPLES

In Figure 10, we provide randomly selected examples of our semantic segmentations after $K$-Means clustering on the PASCAL-VOC dataset. We see that our masks are of similar visual quality to those from MaskContrast (Van Gansbeke et al., 2021), despite the fact that our method is entirely unsupervised.

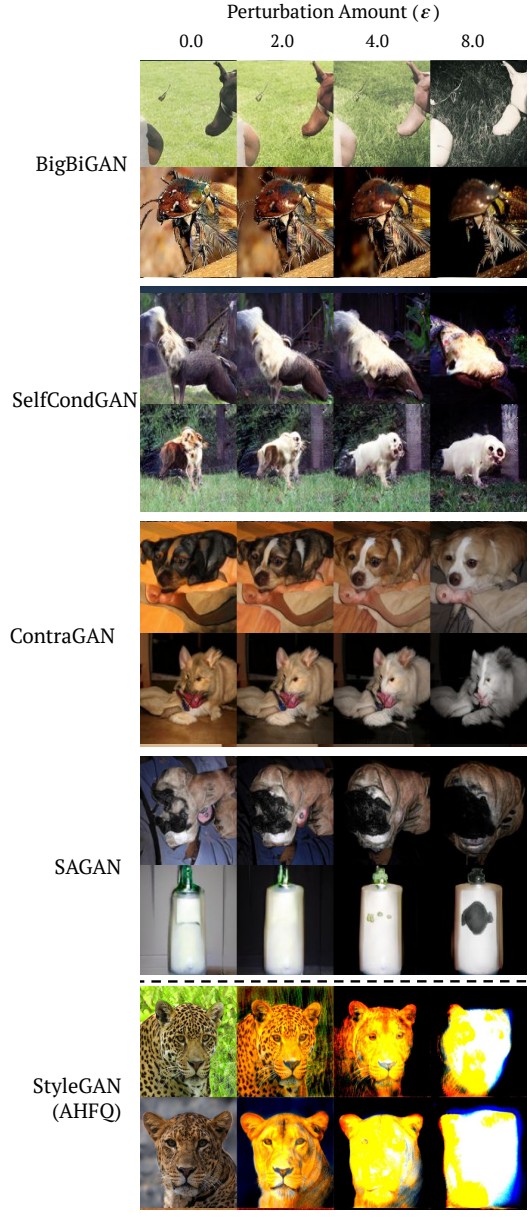

Figure 5: Examples of perturbed images generated by our method for five different generators (GANs). Note that the final generator, StyleGAN (Karras et al., 2019), is only trained on close-up portraits of animals, and thus cannot be used for general-perpose image segmentation. Nonetheless, our method successfully identifies the foreground and background of the generated animal portraits.

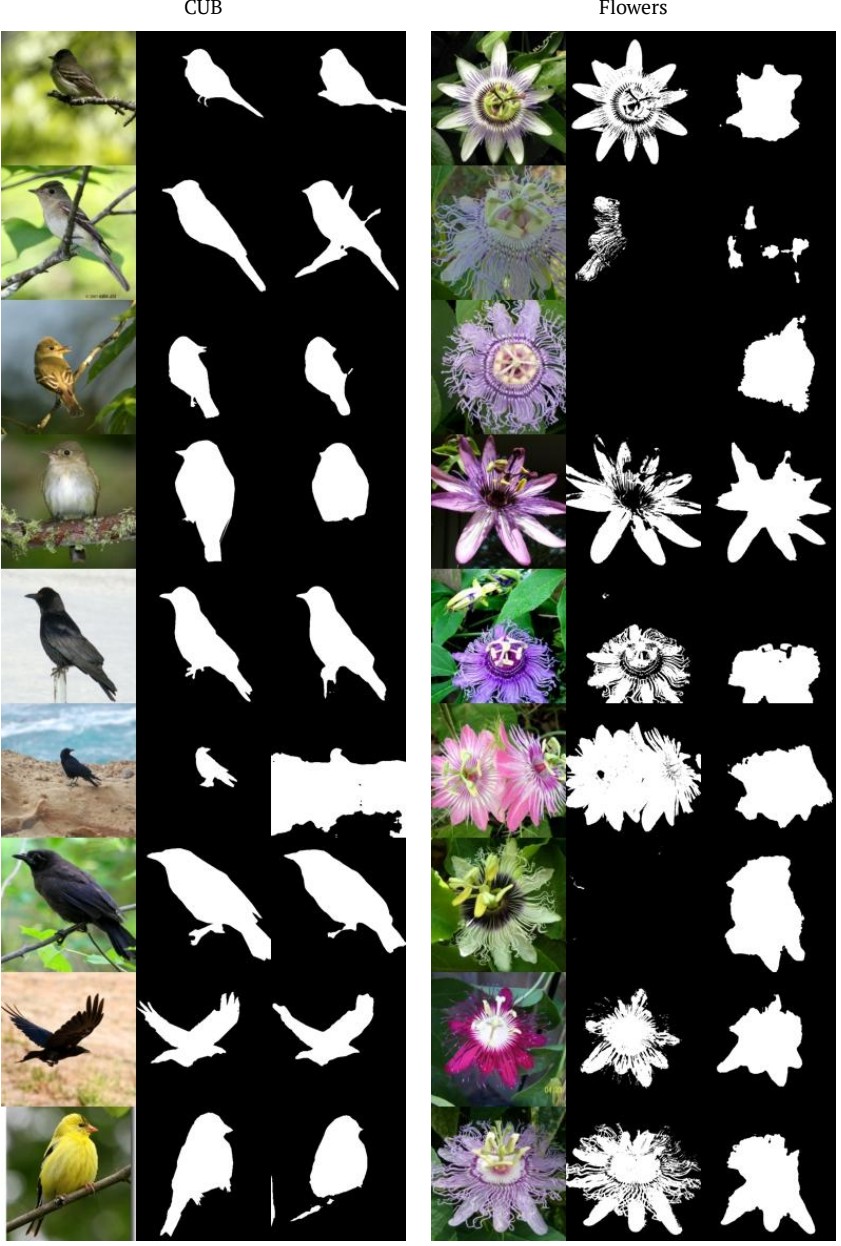

Figure 6: Examples of the final segmentation network across evaluation datasets. From left to right: original image, ground truth, prediction.

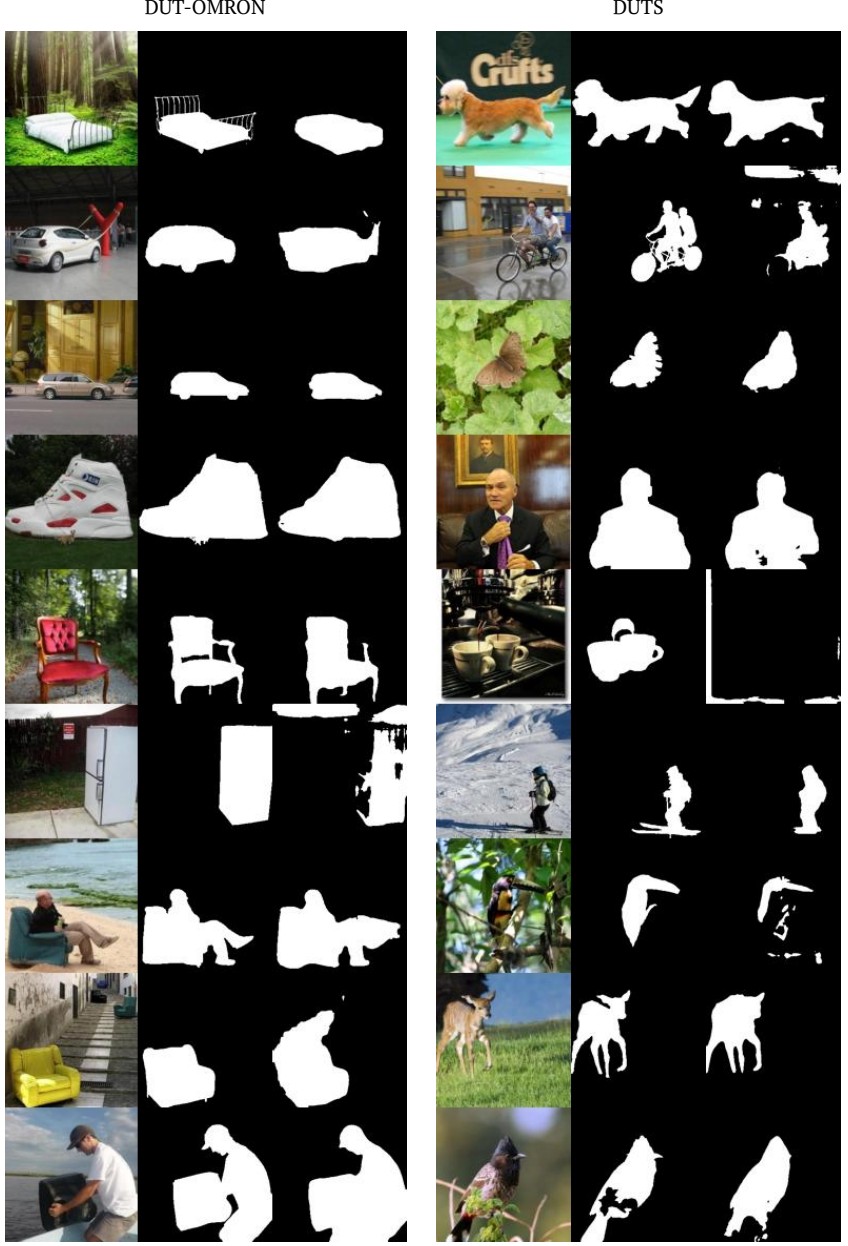

Figure 7: Examples of the final segmentation network across evaluation datasets. From left to right: original image, ground truth, prediction.

ECSSD

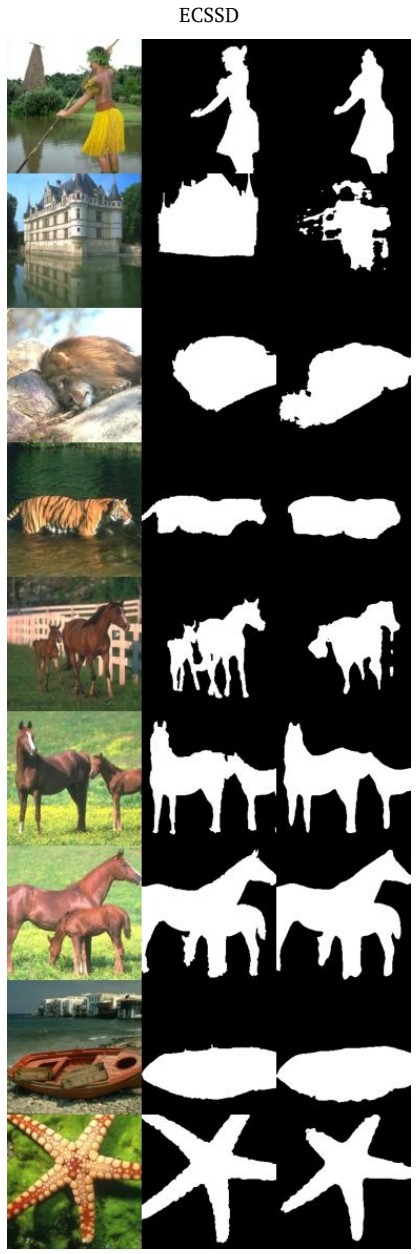

Figure 8: Examples of the final segmentation network across evaluation datasets. From left to right: original image, ground truth, prediction.

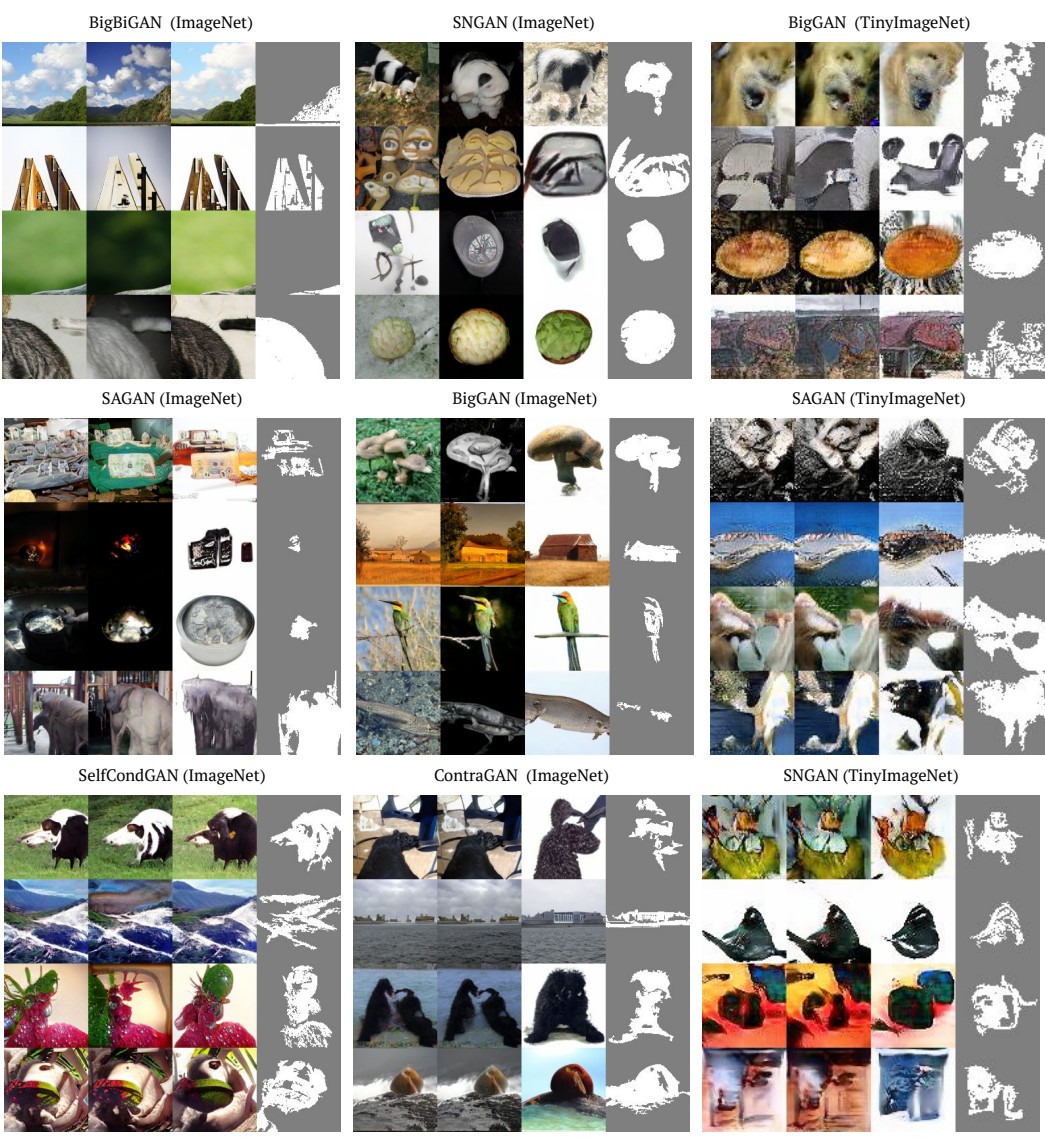

Figure 9: A comparison of perturbed images and their corresponding masks for many different generators.

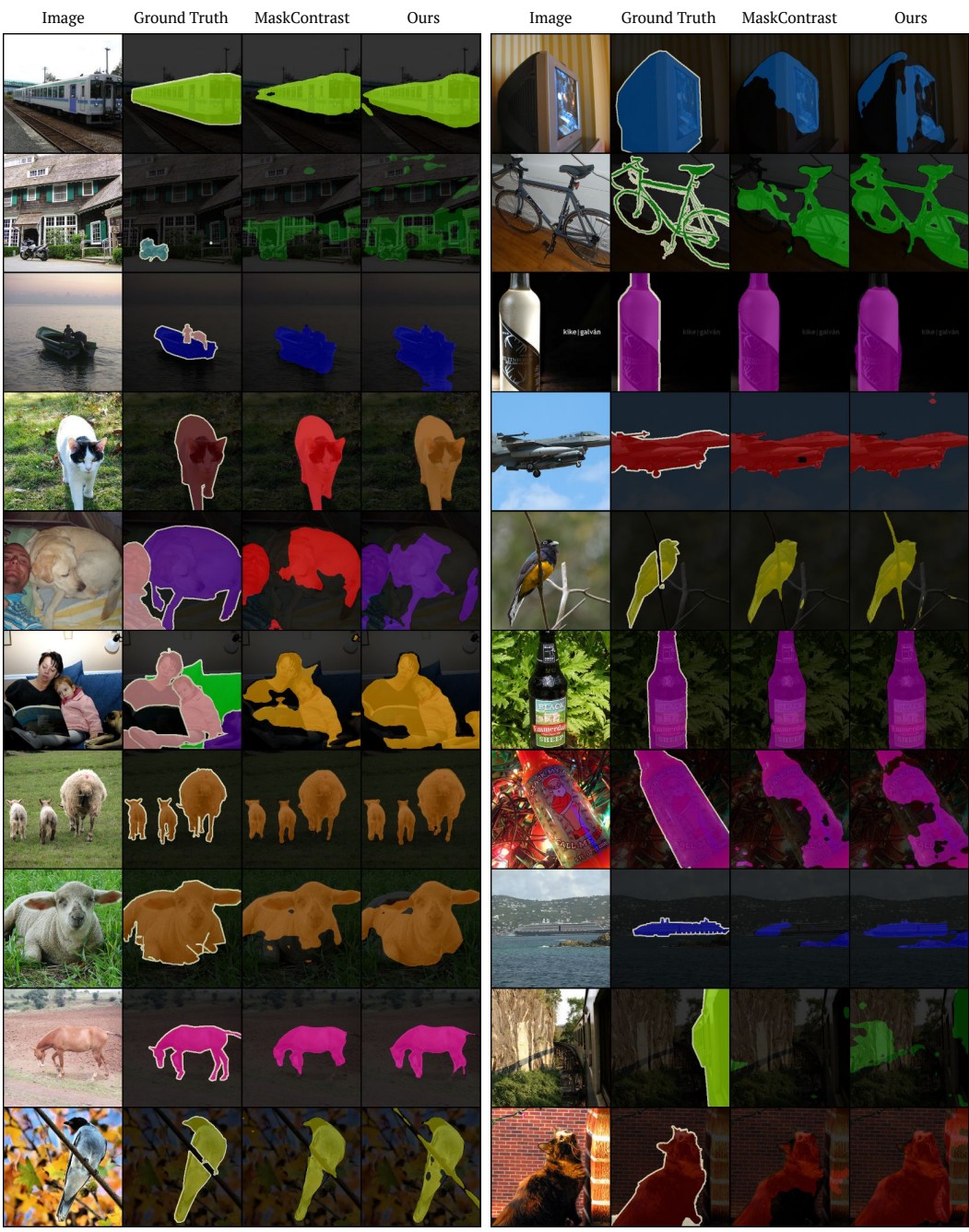

Figure 10: Examples of semantic segmentations after $K$-Means clustering on the PASCAL-VOC dataset. We see that our masks are of similar visual quality to those from MaskContrast (Van Gansbeke et al., 2021), despite the fact that our method is entirely unsupervised. Note that these images are randomly selected, *not* cherry-picked.

# B  FURTHER EXPERIMENTS

Based on the helpful comments of our reviewers, we have conducted new experiments and created more visualizations to better understand how our method functions and performs.

## B.1  ADDITIONAL EXPERIMENTS ON FLOWERS

In this section, we provide a comprehensive explanation and analysis of our performance on the Flowers dataset.

The Flowers dataset is different from the other datasets investigated in our paper (CUB, DUTS, DUT-OMRON, ECSSD, PASCAL VOC) in that the "ground-truth" masks for Flowers were obtained using an automated segmentation method rather than human annotations. As a result, the ground truth are very unreliable. In fact, there are many instances in the Flowers dataset in which the ground truth mask is completely or nearly completely empty despite there clearly being a flower in the image. This issue is also mentioned by Chen et al. (2019), who claim that their method "provide[s] better masks" than the ground-truth in cases of disagreement.

Below, we present an additional experiment to improve our performance on Flowers. We switch the underlying GAN in our method from BigBiGAN (trained on ImageNet) to an unconditional StyleGAN2 trained on Flowers.

This experiment aims to show that our method works well on in-domain data (i.e. using a GAN trained on Flowers to segment Flowers) in addition to out-of-domain data (i.e. using a GAN trained on ImageNet to segment Flowers).

Specifically, we found publicly available weights on GitHub for an (unconditional) StyleGAN2 model Karras et al. (2019) trained on the Flowers dataset at a resolution of 256px, and we applied our method directly to this model. We use perturbation radius $r = 0.2$ rather than $r = 2.0$ as in BigBiGAN because the latent space of StyleGAN2 has a different scale. Apart from this singular parameter, however we apply our method *without any hyperparameter tuning or additional modifications* for the new StyleGAN2.

Results are shown in Table 11. We find that using StyleGAN2 dramatically improves the visual quality of our segmentations both quantitatively and qualitatively. Quantitatively, our performance under this setup is similar to other unsupervised methods which have been extensively finetuned for this task, such as Chen et al. (2019).

We show examples of failure cases in Fig. 13. The most severe failure cases involve empty ground-truth masks. Similarly to ReDo, we find that when our method and the ground truth disagree, our segmentations are often visually superior. At this level of performance, the high level of label noise means that exact mIoU/accuracy numbers are not meaningful. However, there is no doubt that switching generators to StyleGAN tremendously improves our flower segmentation performance.

Additionally, we sought to identify the extent to which this improvement was due purely to the increased resolution (256px) as opposed to the underlying GAN. We train a segmentation model 128px-downsampled versions of the StyleGAN2 generated images and show the results in Table 15. Higher resolution training improves performance modestly on the Flowers dataset, while the bulk of the improvement is due to the use of StyleGAN. This result supports our hypothesis that better GANs correspond to better unsupervised object segmentation performance under our model.

## B.2  ADDITIONAL EXPERIMENTS ON FACES

In this section, we provide additional experiments on the domain of face images.

This section is designed with comparison to Abdal et al. (2021) in mind. However, it is difficult to compare with Abdal et al. (2021) because they evaluate on non-standard datasets, they do not release any information about their data splits, and they do not release code or pretrained models. In particular, they conduct evaluations on CelebA-HQ-Mask using 1000 randomly selected images from the dataset, but they do not specify which images these are. Nevertheless, we tried to recreate their setup as best as possible to establish a comparison.

We applied our method to a StyleGAN2 trained on FFHQ. We use resolution 256px due to time constraints (and 1024px would likely improve results).

We use a perturbation radius $r = 0.2$, which is copied directly from the experiment in the section above (because both experiments use StyleGAN2). Again, with the exception of this single parameter, we *did not change a single other hyperparameter in our entire learning setup* to demonstrate the generalizability of our approach.

Results are shown in Table 12. Qualitatively, we find that our model tends to segment the hair and face but not the clothes, which is an equally valid segmentation but does not align with the ground truth where the person's body is included in the foreground mask. This result could mean that StyleGAN2 internally models the clothing in a different manner from the face which would lead to this behavior in latent space.

### B.3 DATA SIZE

In this section, we investigate the relationship between the amount of synthetic training data and segmentation performance. All experiments in the main paper use 1,000,000 generated examples.

We train models for the same number of total iterations using a varying number of synthetic training images from 100 to 1,000,000. Results are shown in Table 14 and Fig. 11. We find consistently across all datasets that - similar to supervised learning - performance scales logarithmically with the dataset size. Differently from supervised learning, our labels are entirely free to generate, so we can scale our synthetic dataset as large as desired.

### B.4 SEMI-SUPERVISED LEARNING

In this section, we investigate whether our synthetic data can be used to augment existing supervised datasets in a semi-supervised learning setup.

We consider a subset of the CUB dataset with 1000 labeled examples along with synthetic data from our BigBiGAN (ImageNet-pretrained) model. Results are shown in Table 13.

First, we train a supervised model on the 1000 images with ground truth segmentations as a baseline. Similarly, we train a fully unsupervised model on our generated data alone. Now, in the semi-supervised setting, we have access to 1000 labeled images. When training on the combination of both datasets we can already see an improvement over both baselines in IoU (which is the harder and less saturated metric). With access to some in-domain data, we can also take another step to reduce the domain gap between the generated and real images: we can filter our generated images by their distance to the supervised samples. Specifically, we select the 50000 nearest neighbors in our generated set (according to cosine similarity using a self-supervised ResNet-50 He et al. (2019)), and we refer to these as the "kNN generated images" in Table 13.

We see that a model trained on the combined ground truth and kNN samples surpasses both baselines and the supervised model. This result demonstrates that our pipeline can play a valuable role in semi-supervised object segmentation in addition to unsupervised object segmentation.

### B.4.1 FAILURE CASES

In this section, we describe common failure cases of our models.

Fig. 12, Fig. 13, and Fig. 14 show examples of failure cases of our model.

One of the main failure cases of our method is to segment additional foreground objects beyond the "main object" in the image. For example, on the CUB dataset (Fig. 12), we often segment objects such as branches and bird feeders alongside the bird in the image. Across all datasets, one of the other main sources of errors is the object boundary that is not as precise as the object. This could potentially be improved in the future through the use of higher-resolution GANs or by post-processing (e.g. with a CRF).

The Flowers dataset is a special case in which the largest source of error is label noise in the ground truth annotations. Occasionally, our method misses to segment a flower in the background or segments too much of the stem.

With regard to failure cases of mask generation, since there is no ground-truth associated with these generated images, these failure cases were found by manually filtering approximately 500 generated images (Fig. 14). Common failure cases include segmenting too much or too little of the image and are often attributable to low-quality images generated by the GAN. Thus, improvements in the generation quality of GANs and the performance of our method are closely linked.

|  | Data | Acc | IoU |
|---|---|---|---|
| (Chen et al., 2019) | Flowers | 0.879 | 0.764 |
| Ours (BigBiGAN) | ImageNet | 0.796 | 0.541 |
| Ours (StyleGAN) | Flowers | 0.882 | 0.723 |

Table 11: Performance on the Flowers dataset compared to prior methods. It is important to emphasize the fact that the "ground truth" of the Flowers dataset was generated using an automatic procedure and is extremely noisy. For example, a significant fraction of the ground truth masks are entirely empty, and those that are not empty often do not properly reflect the content of the image. As a result, quantitative numbers on the Flowers dataset should be heavily discounted. We encourage the reader to see Fig. 13 for visualizations of failure cases.

|  | Acc | IoU |
|---|---|---|
| Labels4Free (StyleGAN - UNet) | 0.91 | 0.82 |
| Ours (StyleGAN - UNet) | 0.85 | 0.80 |

Table 12: Performance on 1000 random images from the CelebA-HQ-Mask dataset. Performance is not exactly comparable because these models were evaluated using different random subsets.

|  | Real Images | Gen. Images | Acc | IoU |
|---|---|---|---|---|
| Real images only | 1000 | 0 | 0.924 | 0.586 |
| All generated images only | 0 | 1000000 | 0.906 | 0.616 |
| kNN generated images only | 0 | 50000 | 0.908 | 0.588 |
| Combined: real images and all generated images | 1000 | 1000000 | 0.914 | 0.628 |
| Combined: real images and kNN generated images | 1000 | 50000 | **0.931** | **0.665** |

Table 13: Results of combining real images with generated images on the CUB dataset. We use a subset of 1000 training images from CUB along with 1,000,000 generated images. Additionally, we evaluate using a filtered subset of our generated images containing the 50,000 nearest neighbors to the real images. The distance between images was computed using a self-supervised ResNet-50 He et al. (2019). We see that combining real data points with nearby generated data points gives the best results.

| Num. Images | CUB | | Flowers | | DUTS | | OMRON | | ECSSD | |
|---|---|---|---|---|---|---|---|---|---|---|
| | Acc | IoU | Acc | IoU | Acc | IoU | Acc | IoU | Acc | IoU |
| 1000000 | **0.923** | **0.653** | **0.798** | **0.540** | **0.890** | **0.524** | **0.875** | **0.492** | **0.914** | **0.709** |
| 300000 | 0.906 | 0.616 | 0.769 | 0.480 | 0.876 | 0.498 | 0.859 | 0.460 | 0.901 | 0.674 |
| 200000 | 0.903 | 0.606 | 0.784 | 0.508 | 0.868 | 0.473 | 0.852 | 0.440 | 0.898 | 0.659 |
| 100000 | 0.900 | 0.596 | 0.762 | 0.467 | 0.861 | 0.443 | 0.845 | 0.408 | 0.881 | 0.617 |
| 10000 | 0.874 | 0.486 | 0.740 | 0.412 | 0.849 | 0.374 | 0.835 | 0.345 | 0.848 | 0.505 |
| 1000 | 0.847 | 0.419 | 0.700 | 0.343 | 0.821 | 0.311 | 0.811 | 0.299 | 0.802 | 0.410 |
| 100 | 0.752 | 0.256 | 0.650 | 0.284 | 0.727 | 0.203 | 0.728 | 0.206 | 0.710 | 0.271 |

Table 14: Performance of our method for different numbers of generated images. All models are trained for the same number of iterations (20000). For a visual representation of these numbers, see Fig. 11.

| Resolution | Acc | max $F_\beta$ | IoU |
|---|---|---|---|
| 128 | 0.878 | 0.795 | 0.715 |
| 256 | **0.882** | **0.798** | **0.723** |

Table 15: Performance of our method on the Flowers dataset using a StyleGAN 2 model trained on Flowers for different input image resolutions. Using a higher resolution improves performance slightly.

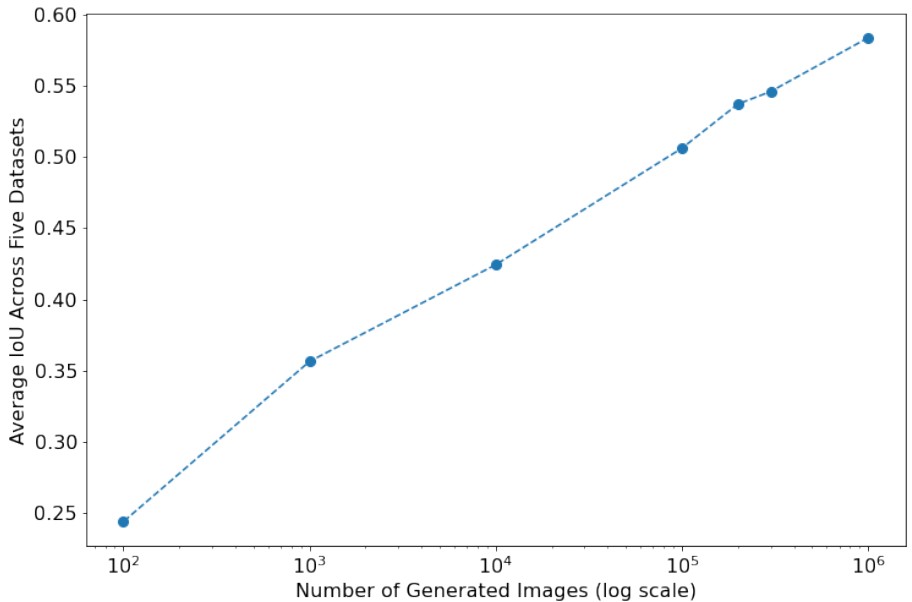

Figure 11: Here we plot the average average mIoU score of a segmentation model as we vary the number of synthetic training images and masks generated using our method. We see that model performance consistently improves with the number of generated training images, following a log-linear scale.

Image Prediction Ground Truth Image Prediction Ground Truth

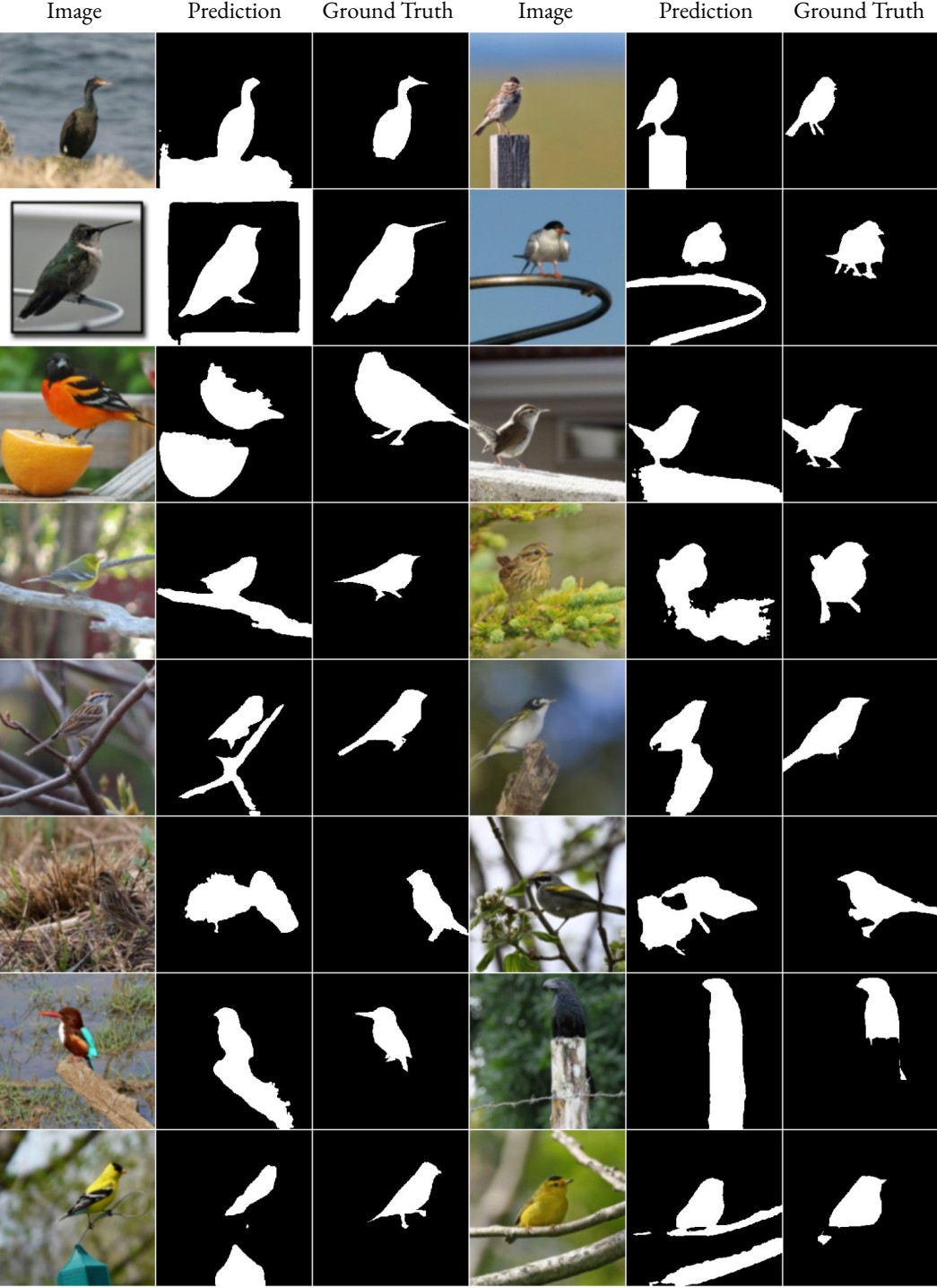

Figure 12: Examples of failure cases of our method on the test set of CUB. Differently from the ground truth, our method frequently segments foreground objects that are not birds, such as branches and bird feeders. This reflects the fact that our model captures general foreground structure in images and has not been trained on CUB.

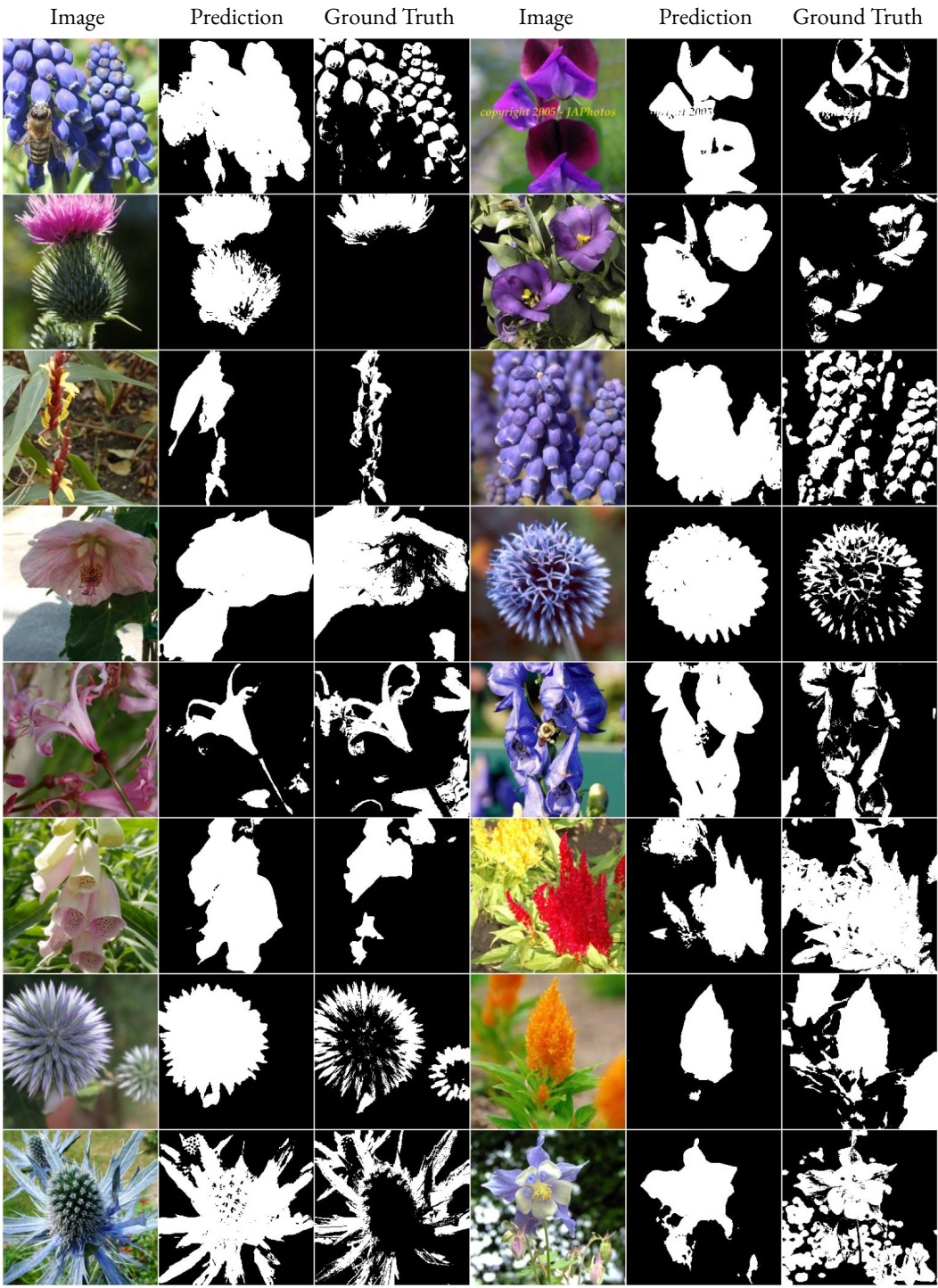

Figure 13: Examples of failure cases on Flowers in which the ground-truth mask is not empty.

Image Mask Image Mask Image Mask

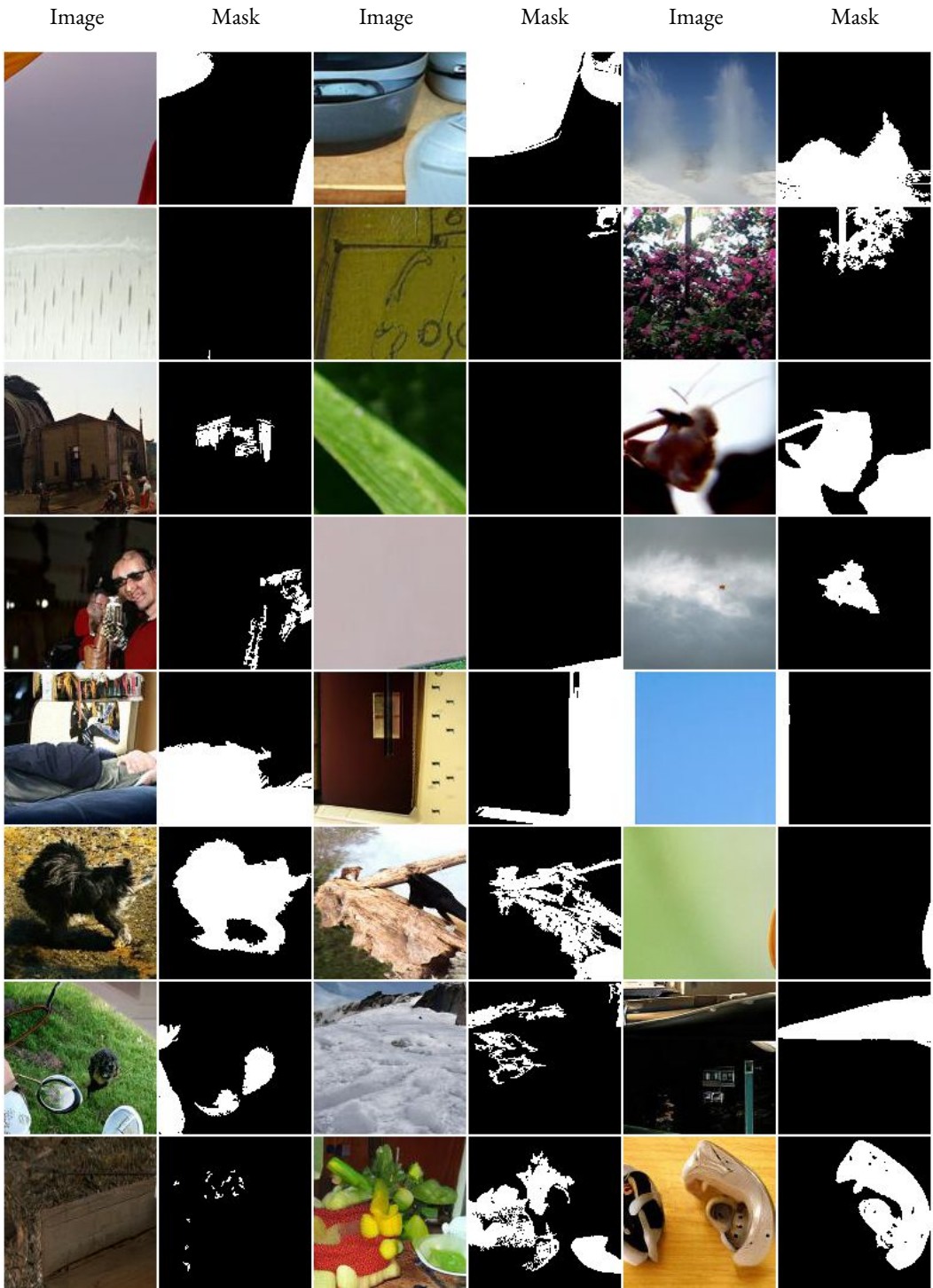

Figure 14: Examples of failure cases for GAN-generated images and masks. As there is no ground-truth associated with these generated images, these failure cases were found by manually filtering approximately 500 generations.

