# OpenReview forum: "Finding an Unsupervised Image Segmenter in each of your Deep Generative Models"
_ICLR.cc/2022/Conference — ICLR 2022 Poster_

### Official Review · Reviewer_zTVT · 2021-10-28

**Correctness:** 3
**Technical Novelty And Significance:** 3
**Empirical Novelty And Significance:** 3
**Recommendation:** 6
**Confidence:** 3

**Main Review:**

This paper proposed to mine useful semantic concepts  from deep generative models which is an interesting and unexplored topic.
Although the core idea of this work is to find plausible directions for latent space manupilation which is similar to [1].  The designed loss functions of edge preserving and image contrast enhancement are reasonable and achieve better image segmentation performance.

However, I still have several concerns.
1. The Sobel Filter is not perfect on extracting the foreground object contour as shown in Fig.2. Therefore, the optimized latent space manipulation direction may not be accurate.
2. The motivation for the image contrast variation is not very plausible. The area covered by the brighreness mask is darked instead of the foreground object which means the mask generation (Eq. 2) not make sense.
3. The authors should add more comparisons with similar approaches such as [2]. I suggest the authors follow the evaluation setting in [2]. The proposed method could synthesize face segmentation samples via StyleGAN2 pretrained on FFHQ. Then train a segmentation model and evaluate it on Celeba-HQ.
4. Most experimental results are based on low resolution images. Could the authors provide more results on high resolutions such as 256x256 or 512x512.
5. The latent manipulation may not be perfect. Please provide some failure cases for mask generation and further discussion.

[1] Andrey Voynov, Stanislav Morozov, and Artem Babenko. Big gans are watching you: Towards unsupervised object segmentation with off-the-shelf generative models. arXiv.cs, abs/2006.04988, 2020.
[2] Rameen Abdal, Peihao Zhu, Niloy Mitra, and Peter Wonka. Labels4free: Unsupervised segmentation using stylegan. 2021.

**Summary Of The Paper:**

This paper presents a new unsupervised algorithm for image segmentation via deep generative models.
Specifically, the authors combined edge preserving and brightness enhancement loss functions to search the latent manipulation directions which only adjusting the foreground object brightness. As a result, the directions could be utilized to synthesize image segmentation training samples.
Experimental results on saliency detection and image segmentation demonstrate the effecteness of the proposed method.

**Summary Of The Review:**

Overall, the reasearch topic is interesting and the proposed method seems to be noval. The experimental evaluation is not very convicing. I may change the rating score If above issues are handled well.

---

> ### Author Response · Authors · 2021-11-22
> **Response 2/2**
>
> > Most experimental results are based on low resolution images. Could the authors provide more results on high resolutions such as 256x256 or 512x512.
>
>
> First, we would like to make sure that the following is clear: although we primarily use low-resolution GAN-generated images for training, all evaluations are carried out on the original full-resolution images.
>
> Second, based on your request and the requests of other reviewers, we conducted a number of additional experiments at 256px. In particular, both the FFHQ experiments above and the Flowers experiments for Reviewer YWmx are performed at 256px.
>
> Third, based on your request, we also trained a model on Flowers at a lower resolution. Here we compare the performance of our model on a resolution of 128 vs. 256 below.
>
> |   Resolution |   Acc |  f_max |   IoU |
> |-------------:|------:|-------:|------:|
> |          128 | 0.878 |  0.796 | 0.715 |
> |          256 | 0.882 |  0.798 | 0.723 |
>
> Although the difference is relatively small, higher resolution training leads to slightly improved performance on the Flowers dataset. Our method is generally independent of the resolution and can scale to higher resolution GANs without changes.
>
> > The latent manipulation may not be perfect. Please provide some failure cases for mask generation and further discussion.
>
>
> We have added Fig. 15 in the appendix to showcase failure cases during mask generation using our latent directions. As there is no ground-truth associated with these generated images, these failure cases were found by manually filtering approximately 500 generated images. Common failure cases are segmenting too much or too little of the image and generally bad samples when the GAN image is of low quality.
>
> Additionally, we discuss segmentation failure cases in Fig 12, 13 and 14 in Appendix B and well as in our response to reviewer JFoE.
>
> Thank you again for your review. We hope that the additional results and explanations have helped answer your questions.

---

> ### Author Response · Authors · 2021-11-22
> **Response 1/2**
>
> We thank the reviewer for their feedback. Below we address their comments regarding the experimental evaluation.
>
> > The Sobel Filter is not perfect on extracting the foreground object contour as shown in Fig.2. Therefore, the optimized latent space manipulation direction may not be accurate.
>
> The search for the latent direction is based on two objectives; with these we wish to alter the appearance of objects without changing their shape. The Sobel operator is responsible for the latter, and even though it might not be perfect at extracting contours, our evaluation suggests that it is sufficient as a structure-preserving constraint. This is necessary so that the produced masks are well-aligned with the entities in the original generated image. We also note that the self-training phase introduces some robustness towards badd training examples. While we agree that our method cannot find _the optimal_ latent direction, it seems impossible to discover it (e.g. a brute-force search in the 120 dimensional latent space is infeasible).
>
> > The motivation for the image contrast variation is not very plausible. The area covered by the brighreness mask is darked instead of the foreground object which means the mask generation (Eq. 2) not make sense.
>
> We do not fully understand this point, but we will do our best to answer. Please let us know if this does not answer your question. We have corrected a typo in Eq. 2, where $v_b \rightarrow v_d$ (please see the revision) and apologize for the misunderstanding. Eq. 2 suggests that we are looking for the difference between the outputs of two directions. We start from an image $x = G(z)$ without perturbations in the latent space ($\epsilon$ = 0) and then perturb along a direction, $v _l=\underset{v}{\operatorname{argmin}}(\lambda L_c(v) + L_s(v))$ with $\lambda=5$, that shifts the foreground to be lighter and the background darker ($G(z + \epsilon v_l)$) and along a direction, $v _d=\underset{v}{\operatorname{argmin}}(\lambda L_c(v) + L_s(v))$ with $\lambda=-5$, that shifts the foreground darker and the background lighter ($G(z + \epsilon v_d)$). As a result, Eq. 2 returns a binary mask with 1 on foreground and -1 on background pixels.
>
> > The authors should add more comparisons with similar approaches such as [2]. I suggest the authors follow the evaluation setting in [2]. The proposed method could synthesize face segmentation samples via StyleGAN2 pretrained on FFHQ. Then train a segmentation model and evaluate it on Celeba-HQ.
>
> We have extensively evaluated our method for 12 GAN models on 5 datasets which are common across related methods. We wish to emphasize, however, that with the exception of (Voynov et al., 2020), all GAN-based methods are learning to segment via layer-wise compositing, which requires reformulating and retraining the GAN on a specific dataset to obtain segmentation masks.
>
> We appreciate your desire to see a comparison with Labels4Free [2]. It is difficult to compare with Labels4Free because they evaluate on non-standard datasets, they do not release any information about their data splits, and they do not release code or pretrained models. In particular, they conduct evaluations on Celeb-A-HQ-Mask using 1000 randomly selected images from the dataset, but they do not specify which images these are. Nevertheless, during the rebuttal period we tried to recreate their setup as best as possible to establish a comparison.
>
> As you requested, we applied our method to a StyleGAN2 trained on FFHQ. We use a model at resolution $256$px due to the limited rebuttal time (and $1024$px would likely improve results). With the exception of the radius of perturbation $r$, which is different for BigBiGAN and StyleGAN2, we \textit{did not change a single other hyperparameter in our entire learning setup} to demonstrate the generalizability of our approach.
>
> We evaluate on a randomly selected set of 1000 masks from the validation set of CelebA-HQ-Mask to try to compare to Labels4Free:
>
> |                               |    Acc |    IoU |
> |:------------------------------|-------:|-------:|
> | Labels4Free (StyleGAN - UNet) |  0.910 | 0.820 |
> | Ours (StyleGAN - UNet)        |  0.850 | 0.800 |
>
> Qualitatively, we find that our model tends to segment the hair and face but not the clothes, which is an equally valid segmentation but does not align with the ground truth where the person’s body is included in the foreground mask. This result could mean that StyleGAN2 internally models the clothing in a different manner from the face which would lead to this behavior in latent space.

---

### Official Review · Reviewer_JFoE · 2021-10-28

**Correctness:** 3
**Technical Novelty And Significance:** 2
**Empirical Novelty And Significance:** 2
**Recommendation:** 5
**Confidence:** 3

**Main Review:**

Strengths:
- The approach is well-described and the paper is generally easy to follow.
- The literature review is comprehensive.
- The approach seems to produce good qualitative results, and quantitative results are slightly better than competing unsupervised approaches.
- Quantitative results are computed on a variety of tasks, not just instance segmentation.

Weaknesses:
- In my opinion, there is little novelty. Except for the latent direction discovery step, the overall framework is very similar to existing approaches.
- The latent direction is found by exploiting the photographer's bias (i.e. objects usually appear at the center of the image). How does the method perform if this condition is not satisfied?
- There is no discussion of failure cases (I think they would make the paper stronger).

**Summary Of The Paper:**

The paper proposes an unsupervised approach for instance segmentation that leverages pretrained GANs. The approach is divided into two steps: first, the authors discover a latent direction in the generator that allows them to segment generated images. Then, the resulting image-label pairs are used to train a segmentation network. This network can finally be used to perform inference on unseen images. The approach is evaluated on a variety of GAN architectures. The main contribution claimed by the authors is that their method can be applied to off-the-shelf pretrained GANs, whereas competing approaches require architectural variations and re-training.

**Summary Of The Review:**

Although the paper is well-written and its implementation seems to be technically sound, my main concern is the limited novelty of the proposed approach. There are already many existing approaches that leverage GANs in order to segment images, and while this approach presents some useful ideas, I think they are somewhat incremental. Overall, the paper seems to be below the bar, but I would also like to hear the author's opinion and the other reviewers.

---

> ### Author Response · Authors · 2021-11-22
> **Response**
>
> We thank the reviewer for the constructive feedback.
>
> > In my opinion, there is little novelty. Except for the latent direction discovery step, the overall framework is very similar to existing approaches.
>
> Our approach differs from other approaches in that we seek a general method to find foreground/background structure implicitly encoded in standard, non-layerwise GANs rather than encoding it explicitly (by building a layerwise GAN framework) or searching for it manually. This enables us to leverage any of the numerous existing generators that have already been pretrained on millions of high resolution images. Differently from layer-wise approaches, our approach does not require training new GANs and it does not rely on the assumption that the foreground and background of an image are independent. As such, we are able to show that our method is applicable over a number of different GAN models without any modifications and show that segmentation performance is correlated to generation quality, which contributes to the field of interpretability as well as the field of unsupervised segmentation.
>
> > The latent direction is found by exploiting the photographer's bias (i.e. objects usually appear at the center of the image). How does the method perform if this condition is not satisfied?
>
> Thank you for bringing up this point. We actually answered exactly this question in an experiment in the appendix of the paper (A.3 and Table 10), which is referred to on page 8 in the section “Ablation: Varying λ, , the Central Prior, Random Initializations.” We realize now that this result should have been displayed more prominently, and we have updated the paper accordingly.
>
>
> For simplicity, we summarize the experiment here. This experiment does not use a central prior and replaces it with a variance term: the variance in brightness across the image is maximized instead. This prior has no spatial component and thus is not affected by the location of objects in the image.
>
> |      |                 | CUB   | Flowers | DUT-O. | DUTS  | ECSSD |
> |------|-----------------|-------|---------|--------|-------|-------|
> | Acc. | full model      | 0.912 | 0.773   | 0.878  | 0.890 | 0.905 |
> |      | no center prior | 0.885 | 0.788   | 0.825  | 0.849 | 0.876 |
> | IoU  | full model      | 0.601 | 0.479   | 0.451  | 0.486 | 0.663 |
> |      | no center prior | 0.535 | 0.553   | 0.379  | 0.416 | 0.609 |
>
> We see a small performance drop relative to the central prior, but we are still able to successfully segment objects without supervision.
>
> > There is no discussion of failure cases (I think they would make the paper stronger).
>
> Thank you for the suggestion. We have added a discussion of failure cases to appendix B (Figure 12, 13 & 14).
> The main findings are:
> * One of the main failure cases of our method is to segment additional foreground objects beyond the “main object” in the image. For example, on CUB (Fig. 12), we often segment objects such as branches and bird feeders alongside the bird in the image.
> * Across all datasets, one of the other main sources of errors is the object boundary that is not as precise as the object. This could potentially be improved in the future through the use of higher-resolution GANs or by post-processing (e.g. with a CRF).
> * On the Flowers dataset, the largest source of error is label noise in the ground truth annotations. We discuss this issue extensively in our response to Reviewer YWmx. Fig. 13 shows our top 16 predictions by mIoU error in the Flowers dataset: the ground truth segmentations are empty. Finally in Fig 14, we thus show top failure cases on Flowers where the ground truth mask is not empty, which still mainly consists of low quality ground truth annotations. Occasionally, our method misses to segment a flower in the background or segments too much of the stem.

---

### Official Review · Reviewer_XSNi · 2021-11-02

**Correctness:** 3
**Technical Novelty And Significance:** 3
**Empirical Novelty And Significance:** 2
**Recommendation:** 6
**Confidence:** 3

**Main Review:**

The motivation of the paper and the overall method is very interesting. The authors prove that by using their method, they can leverage and find foreground-background information in virtually any GAN architecture and use those cues to create a synthetic dataset that is then used for image segmentation. I think that this has a lot of applications and is an interesting area of research.

The paper is fairly well written and the overall message is well understood. I think that the authors demonstrate that their method can be applied to various architectures and that overall it has potential on identifying the foreground-background information.

However, I find the rest of the experimental part relatively weak. More exactly, while it is interesting to find that the method works with 12 other GANs, I feel like that for the image segmentation part there are some missing key aspects. The first one, is the comparison with methods that do not use/are not generative. While I understand the motivation and clearly that the goal is to directly compare agains generative methods, I think that a comparison with all methods is needed. This will give a clear indication on how the generative methods compare against other methods.

Moreover, I feel like a lot more details regarding what data they use for training the models is missing. In order to understand the power of this generated data, I feel like the authors should have answered to the following questions, but in the current form of the paper are missing:
1) How many samples does the final dataset contain?
2) How does the number of samples affect the performance of the final model?
3) What happens if you combine data from multiple GANs?

Lastly, since the overall performance is relatively on par with other methods, while having some advantages in terms of supervision, I would have wished to see at least one experiment that augments existing datasets with synthetic data. In this form, while I like the idea, the numbers do not seem that convincing to me.

**Summary Of The Paper:**

The paper tackles the problem of image segmentation. It focuses on finding foreground-background image separation cues from a pre-trained GAN and create a synthetic dataset. Once the dataset is created, it trains a segmentation models and test the approach on several benchmarks. The paper is centred around how to extract and detect this foreground-background separation in an unsupervised way. The method works with any GAN architecture and in the end achieves competitive results while being unsupervised.

**Summary Of The Review:**

Overall I think that the paper is well written. My major concerns when it comes to the decision are related to the comparison with non generative methods and the lack of insights and information over the data used for training. Please see above for more details.

---

> ### Author Response · Authors · 2021-11-22
> **Response**
>
> We thank the reviewer for their constructive feedback and suggestions for improving the experimental part of our method.
>
> > How many samples does the final dataset contain? How does the number of samples affect the performance of the final model?
>
> The dataset for experiments in the paper contains one million samples. We have added this to the implementation details in the revised appendix.
>
> Next, we ran an experiment in which we varied the number generated samples (and kept the number of iterations constant):
>
> |             | CUB   |       | Flowers |       | DUTS  |       | DUT_OMRON |       | ECSSD |       |
> |-------------|-------|-------|---------|-------|-------|-------|-----------|-------|-------|-------|
> | Num. Images | Acc   | IoU   | Acc     | IoU   | Acc   | IoU   | Acc       | IoU   | Acc   | IoU   |
> | 1000000     | 0.923 | 0.653 |  0.798  | 0.540 | 0.890 | 0.524 |   0.875   | 0.492 | 0.914 | 0.709 |
> | 300000      | 0.906 | 0.616 |  0.769  | 0.480 | 0.876 | 0.498 |   0.859   | 0.460 | 0.901 | 0.674 |
> | 200000      | 0.903 | 0.606 |  0.784  | 0.508 | 0.868 | 0.473 |   0.852   | 0.440 | 0.898 | 0.659 |
> | 100000      | 0.900 | 0.596 |  0.762  | 0.467 | 0.861 | 0.443 |   0.845   | 0.408 | 0.881 | 0.617 |
> | 10000       | 0.874 | 0.486 |  0.740  | 0.412 | 0.849 | 0.374 |   0.835   | 0.345 | 0.848 | 0.505 |
> | 1000        | 0.847 | 0.419 |  0.700  | 0.343 | 0.821 | 0.311 |   0.811   | 0.299 | 0.802 | 0.410 |
> | 100         | 0.752 | 0.256 |  0.650  | 0.284 | 0.727 | 0.203 |   0.728   | 0.206 | 0.710 | 0.271 |
>
> We also show this information in graphical form in Figure 11 (on page 23) of the updated paper.
>
> We find consistently across all datasets that - similar to supervised learning - performance scales logarithmically with the dataset size. Differently from supervised learning, our labels are entirely free to generate, so we can scale our synthetic dataset as large as desired.
>
> > What happens if you combine data from multiple GANs?
>
> We have not yet experimented with combining data from multiple GANs, but it is a very interesting question. Given the results of the previous experiment and the experiments we conducted for Reviewer YWmx, our intuition is that we would see improvements in performance when combining multiple GANs. We suspect that these improvements would result from the increased diversity in generated images and the fact that different models may have different biases with regard to mask generation.
>
> We will conduct an experiment on combining data from different GANs and add it to the final version.
>
> > Augment existing datasets with synthetic data.
>
> Thank you for this suggestion. We have run the following experiments (described below):
>
> |                                                |      Acc |      IoU |
> |:-----------------------------------------------|---------:|---------:|
> | Real images only                               | 0.924     | 0.586    |
> | All generated images only                      | 0.906     | 0.616    |
> | kNN generated images only                      | 0.908     | 0.588    |
> | Combined: real images and all generated images | 0.914     | 0.628    |
> | Combined: real images and kNN generated images | 0.931     | 0.665    |
>
> On the CUB dataset, we train a supervised model on 1000 images with ground truth segmentations as a baseline. Similarly, we train a fully unsupervised model on the generated data alone. Now, in the semi-supervised setting, we have access to 1000 labeled images. When training on the combination of both datasets we can already see an improvement over both baselines in IoU (which is the harder and less saturated metric). With access to some in-domain data, we can also take another step to reduce the domain gap between the generated and real images: we can filter our generated images by their distance to the supervised samples. Specifically, we select the 50000 nearest neighbors in our generated set (according to cosine similarity using a self-supervised ResNet-50), and we refer to these as the “kNN generated images” in the table above.
>
> We see that a model trained on the combined ground truth and kNN samples surpasses both baselines and the supervised model.
>
> We have also added this new experiment and their discussion to the appendix of the revised paper.
>
> > Comparison with non-generative models
>
> This seems to be a misunderstanding. We do indeed compare with non-generative models. However, these approaches generally do not perform well on our tasks. For example, IIC which is based on clustering, falls behind many previous methods (Tab 1).
> Discriminative approaches are often geared toward semantic segmentation rather than object segmentation. As a result, we compare to them extensively in that setting (Tab 3). Colorization, IIC, and MaskContrast are discriminative methods. Please let us know if we have missed a discriminative method in our comparison.
>
> Thank you again for your review. We hope that these experiments have answered your questions.

---

### Official Review · Reviewer_YWmx · 2021-11-05

**Correctness:** 4
**Technical Novelty And Significance:** 4
**Empirical Novelty And Significance:** 3
**Recommendation:** 8
**Confidence:** 4

**Main Review:**

=== PROS ===
+ This paper tackles an interesting problem : that of forging an image segmentation model without supervision. It does so in an interesting way, by exploiting a latent structure in image generation models. While the motivation in intro and p4 is a bit handwavy, it seems to be verified by experiments.
+ The proposed approach is very simple and can be adapted to any generative model. The experiments further show that the performance of the segmentation is correlated with the PID, a proxy measure for the quality of generated images.

=== CONS ===
- The performance on the Flowers dataset is a bit disappointing w.r.t. to other unsupervised methods. Could it be because of the domain on which the initial generative model was trained on? If the generative model was specialized in flower species, would the results change?

**Summary Of The Paper:**

This paper describes a simple procedure that allows to train an unsupervised image segmentation model only using a generative model as input. By tricking the generator to predict proxy segmentation masks along with a generated image, the method allows to construct a possibly infinite dataset of images associated with a F/B mask. A segmentation model can then be trained on this dataset to predict foreground.
The proposed approach is evaluated across a wide range of small-scale segmentation datasets, and compared with various approaches to the problem (both supervised, handcrafted and unsupervised). An adaptation of this approach to semantic segmentation is proposed and evaluated on Pascal VOC.

**Summary Of The Review:**

Overall, I think that this is an interesting paper, showing that generative models contain a lot of information about the structure of images. Being able to build a segmentation model that really works on real data is quite impressive. As stated by the authors, this indeed is an interesting application of generative models that is useful on actual CV tasks. Because of the surprising observation and decent experimental execution, I think this paper deserves to be presented at ICLR.

---

> ### Author Response · Authors · 2021-11-22
> **Response**
>
> We thank the reviewer for the positive feedback.
>
> > The performance on the Flowers dataset is a bit disappointing w.r.t. to other unsupervised methods. Could it be because of the domain on which the initial generative model was trained on? If the generative model was specialized in flower species, would the results change?
>
> Thank you for pointing out the anomalously low performance of our method on the Flowers dataset. We refrained from fully explaining the situation with this dataset in the paper, but we now realize that we should have included a full explanation.
>
> The Flowers dataset is different from the other datasets investigated here in that the “ground-truth” masks for Flowers were obtained using an automated segmentation method and are very unreliable. In fact, there are many instances in the Flowers dataset in which the ground truth mask is completely or nearly completely empty despite there clearly being a flower in the image. This issue is also mentioned by the ReDo paper, which claims that their method “provide[s] better masks” than the ground-truth in cases of disagreement.
>
> Nevertheless, motivated by your comment, we conducted an experiment to improve our performance on Flowers: we switch the underlying GAN in our method from BigBiGAN (trained on ImageNet) to an unconditional StyleGAN2 trained on Flowers. This switch makes sense because flowers compose only a very small fraction of ImageNet, so leveraging in-domain data could improve results.
>
> Specifically, we found publicly available weights on GitHub for an (unconditional) StyleGAN-2 model trained on the Flowers dataset at a resolution of $256$px, and we applied our method directly to this model. With the exception of the perturbation radius $r$, which is different for BigBiGAN and StyleGAN, we use our method \textit{without any hyperparameter tuning or additional modifications} for the new StyleGAN.
>
> Full experimental details and discussion have been added to the appendix of the revised paper. In summary, we find that using StyleGAN 2 dramatically improves the visual quality of our segmentations.
>
> Quantitatively, we now achieve similar performance to other unsupervised methods that you mention in your review, such as ReDo (shown below):
>
> |        | Data               |   Acc |   IoU |
> |:-------|:-------------------|------:|------:|
> | ReDo   | Flowers            | 0.879 | 0.764 |
> | Ours   | ImageNet           | 0.796 | 0.541 |
> | Ours   | Flowers            | 0.882 | 0.723 |
>
> We show examples of failure cases in Figures 13 and 14 of the updated paper. The most severe failure cases involve empty ground-truth masks. Similarly to ReDo, we find that when our method and the ground truth disagree, our segmentations are often visually superior. At this level of performance, the high level of label noise means that exact mIoU/accuracy numbers are not meaningful. However, there is no doubt that switching generators to StyleGAN tremendously improves our flower segmentation performance.
>
> We hope that this response clarifies the Flowers dataset situation and answers your question.
>
> Thank you once again for your review!

---

### Public Comment · ~Andrey_Voynov1 · 2021-11-12
**Note on an Updated version of a cited paper**

Dear authors,

First I would like to thank you for sharing the paper, it was interesting to read it!

Let me note our recent update of the "Big gans are watching you: Towards unsupervised object segmentation with off-the-shelf generative models" paper. We have updated the arXiv version: https://arxiv.org/abs/2006.04988 and the paper was published on ICML'21 as "Object Segmentation Without Labels with Large-Scale Generative Models" http://proceedings.mlr.press/v139/voynov21a/voynov21a.pdf
The major update introduced in that revision was a fully unsupervised algorithm to distinguish the background-foreground latent direction (sections 3.1 - 3.2) thus we think that our approach should be treated as unsupervised instead of weakly-supervised.

Best regards, Andrey Voynov

---

> ### Author Response · Authors · 2021-11-22
> **Thank you.**
>
> Dear Dr. Voynov,
>
> We are very glad to hear that you enjoyed reading our paper!
>
> We also would like to thank you for making us aware of the updated version of your paper, which we had not seen due to its different title. We have updated our paper to include your method in the “Unsupervised” category.
>
> Best regards,
> (Anonymous Authors)

---

### Author Response · Authors · 2021-11-22
**Revision Summary**

Dear Reviewers,

Thank you for your time and the constructive reviews. We appreciate your thorough readings of the paper and your insightful questions. Based on these questions, we have added additional discussions to our paper and conducted five new experiments to further demonstrate the usefulness of our method.

We have responded to the individual comments below. We have also updated the paper correspondingly and marked changes with red font.

The new results and analysis prompted by your comments are placed in a new section of the appendix (Appendix B) on pages 25-33.

Main changes:
* We have run five additional experiments based on the reviewers’ suggestions. The results are consistent with the messages of the paper and we believe that they answer the raised questions. All results and their discussion can be found in Appendix B and the individual replies.
* We have included an additional discussion of failure cases in the appendix.
* We have updated the claims and Table 1 with respect to Voynov et al. (2020) based on their comment here (the method is now referred to as unsupervised instead of weakly supervised).
* We have fixed two typos in Section 3 and added a reference to Table 10 in the main body of the paper.

---

### Decision · Program_Chairs · 2022-01-20

**Decision:**

Accept (Poster)

**Comment:**

The reviewers all agree that this paper proposes a very interesting approach of finding useful information encoded inside a generative model. They show how foreground/background semantics learnt in a generative model are useful for tasks like segmentation.
This is a general approach that can be applied to other models in the future.
It is an accept.